JCB Journal of Cell Biology

**TOOLS**

# Engineered SUMO/protease system identifies Pdr6 as a bidirectional nuclear transport receptor

Arturo Vera Rodriguez, Steffen Frey, and Dirk Görlich

Cleavage of affinity tags by specific proteases can be exploited for highly selective affinity chromatography. The SUMO/SENP1 system is the most efficient for such application but fails in eukaryotic expression because it cross-reacts with endogenous proteases. Using a novel selection system, we have evolved the SUMO[Eu]/SENP1[Eu] pair to orthogonality with the yeast and animal enzymes. SUMO[Eu] fusions therefore remain stable in eukaryotic cells. Likewise, overexpressing a SENP1[Eu] protease is nontoxic in yeast. We have used the SUMO[Eu] system in an affinity-capture-proteolytic-release approach to identify interactors of the yeast importin Pdr6/Kap122. This revealed not only further nuclear import substrates such as Ubc9, but also Pil1, Lsp1, eIF5A, and eEF2 as RanGTP-dependent binders and thus as export cargoes. We confirmed that Pdr6 functions as an exportin in vivo and depletes eIF5A and eEF2 from cell nuclei. Thus, Pdr6 is a bidirectional nuclear transport receptor (i.e., a biportin) that shuttles distinct sets of cargoes in opposite directions.

## Introduction

Cell nuclei rely on protein import from the cytoplasm while producing and exporting, e.g., ribosomes or mRNAs. This exchange occurs through nuclear pore complexes (NPCs)—giant molecular machines with an elaborate modular structure (reviewed by Hoelz et al., 2016; Schwartz, 2016). Intrinsically disordered FG repeat domains control this nucleocytoplasmic transport; they condense into a dense "FG phase" that serves as the permeability barrier within the central NPC channel (reviewed by Schmidt and Görlich, 2016). The barrier is highly permeable for small molecules, but usually retains larger species with a size limit of ~5 nm or ~30 kD. This retention allows the cell nucleus and cytoplasm to keep different sets of proteins and thus be specialized in function.

Nuclear transport receptors (NTRs) of the importin β superfamily can overcome the aforementioned size limit. They translocate in a facilitated manner through the barrier, draw energy from the RanGTPase system, and pump cargoes through NPCs. According to the direction of cargo transport, NTRs can be grouped into importins and exportins (reviewed by Güttler and Görlich, 2011; Christie et al., 2016; Baumhardt and Chook, 2018).

Importins recruit their cargoes in the cytoplasm, enter nuclei, and release cargo upon encountering RanGTP. They return as importin–RanGTP complexes to the cytoplasm, where GTP hydrolysis disengages Ran, allowing the importins to bind and import another cargo molecule. Exportins operate the opposite way. They require nuclear RanGTP for cargo binding, translocate as cargo–exportin–RanGTP complexes to the cytoplasm, and then release the cargo upon GTP hydrolysis.

Exploring the cargo spectrum of individual NTRs is still a major effort in the field, with recent studies aiming at comprehensive transport substrate assignments (Kirli et al., 2015; Kimura et al., 2017; Mackmull et al., 2017; Baade et al., 2018). The literature so far suggests that the vast majority of NTRs function either as an importin or an exportin. Nevertheless, 3 of the 19 mammalian NTRs (importin 13, Xpo4, and Xpo7) and 1 of the 14 yeast NTRs (Msn5) were reported to carry distinct sets of cargoes in opposite directions (Kaffman et al., 1998; Lipowsky et al., 2000; Mingot et al., 2001, 2004; Yoshida and Blobel, 2001; Gontan et al., 2009; Aksu et al., 2018). Given these small numbers, it has been commonly assumed that bidirectional transport is an exception rather than the rule.

*Saccharomyces cerevisiae* Pdr6/Kap122 is a so far underexplored NTR with just two import substrates being reported, namely, the Toa1/Toa2 subunits of TFIIA and Wtm1 functioning apparently as an import adapter for ribonucleotide reductase (Titov and Blobel, 1999; Zhang et al., 2006). In this study, we revisited the cargo spectrum of Pdr6 and exploited a newly engineered SUMO/SUMO-protease system for this purpose.

SUMO (Matunis et al., 1996; Mahajan et al., 1997) is conjugated to a large number (perhaps thousands) of proteins and has a great impact on cellular physiology (reviewed by Flotho and Melchior, 2013; Hendriks and Vertegaal, 2016).

Department of Cellular Logistics, Max Planck Institute for Biophysical Chemistry, Göttingen, Germany.

Correspondence to Dirk Görlich: goerlich@mpibpc.mpg.de; S. Frey's present address is NanoTag Biotechnologies GmbH, Göttingen, Germany.

SUMO proteases/isopeptidases mediate de-conjugation (Li and Hochstrasser, 1999). Furthermore, they process the C-terminus of newly made SUMO molecules to expose the characteristic Gly-Gly C-terminal motif; this makes SUMO conjugatable in the first place.

SUMO (and other ubiquitin-like modifiers) are widely used as fusion tags in recombinant protein expression, typically in combination with N-terminal polyhistidine or alternative affinity tags that confer a selective binding to a cognate affinity matrix. SUMO tags are cleavable by a SUMO protease, which allows tag removal either as a post-purification treatment (Malakhov et al., 2004) or even as a highly selective elution step from the affinity matrix (Frey and Görlich, 2014a,b). Advanced options are double-tag purifications, where two subunits of a protein complex are fused with orthogonal affinity tags and protease cleavage sites (PCSs).

Compared with other tag-cleaving proteases (i.e., TEV-protease, thrombin, factor Xa, enterokinase, and human rhinovirus 3C protease), SUMO proteases feature a unique combination of advantages (Malakhov et al., 2004; Frey and Görlich, 2014a): they are extremely active and highly specific, they cleave robustly in a wide range of buffers, and they leave no undesired residues at the P1′ position. Furthermore, the so far used SUMO proteases are straightforward to produce, and the SUMO fusion module typically enhances protein expression and stability.

A drawback of the SUMO technology is that endogenous SUMO proteases prematurely cleave SUMO tags in eukaryotic cells, which restricts its use to prokaryotic hosts such as Escherichia coli. A yeast SUMO mutant named SUMOstar is so far the only solution to this problem (Peroutka et al., 2008). SUMOstar is stabilized by two point mutations against cleavage by WT SUMO proteases but cleavable by an engineered SUMOstar protease of relaxed specificity. Nevertheless, a single PCS is not yet sufficient for advanced multi-tag purifications in eukaryotic hosts, and we found that SUMOstar fusions are not fully stable when expressed in yeast.

As an alternative and improved solution, we evolved a new set of SUMO$^{Eu}$ variants that are highly resistant against SUMOstar or WT SUMO proteases. They behave as stable fusion tags in yeast and human cells. Moreover, we engineered a set of SENP$^{Eu}$ proteases that cleave SUMO$^{Eu}$ fusions with great specificity and very high turnover. We demonstrate the superior utility of the SUMO$^{Eu}$ system for recombinant protein expression and purification in S. cerevisiae and in human HEK-293T cells. Furthermore, we used the new affinity purification system to revisit the cargo spectrum of the yeast importin Pdr6 and identified the SUMO-conjugating enzyme Ubc9 as an additional import substrate. Strikingly, however, we also identified and validated four export cargoes, namely, the two BAR domain proteins Pil1 and Lsp1 as well as the translation factors eIF5A and eEF2, which apparently leak into nuclei and then require Pdr6 for retrieval to the cytoplasm. This establishes Pdr6 as an exportin and suggests that a bidirectional mode of operation is far more common than previously thought. Accordingly, we propose the term "biportin" to describe this type of NTRs.

## Results

### A system for evolving proteases to novel specificities

SUMO proteases are very useful for cleaving tags from recombinant (SUMO-tagged) proteins and enhancing the specificity of purifications by the affinity capture and proteolytic release strategy. This is straightforward when proteins are produced in E. coli but doomed to fail in eukaryotic protein expression systems where endogenous SUMO proteases cleave such tags prematurely. This can be solved by mutagenizing SUMO to resist cleavage by the interfering proteases and then evolving a protease that accepts the new SUMO variant as a substrate again.

To find a general solution to such an evolution problem, we designed an in vivo selection system, where E. coli cells express a protease variant from a first plasmid, while a second plasmid encodes a modular "protease-selectivity sensor." The sensor links two PCSs with two degradation signals (degrons) and an antibiotic-resistance marker in such a way that resistance can only occur when the protease cleaves the desired site (PCS$^{For}$) but leaves the other site (PCS$^{Against}$) intact (Fig. 1 A).

The N-terminal sensor module includes a candidate cleavage site (PCS$^{Against}$) followed by an N-end rule degron (Bachmair et al., 1986), which remains silent in the fusion context. Cleavage of PCS$^{Against}$ exposes a destabilizing N-terminal residue, activates this degron, and leads to degradation of the fused antibiotic resistance protein and thus to death under selective conditions.

The C-terminal part of the fused sensor comprises the other candidate cleavage site (PCS$^{For}$) followed by the ssrA degradation signal (reviewed by Keiler, 2008; Himeno et al., 2014), which, by default, triggers degradation and hence loss of antibiotic resistance. Cleavage of PCS$^{For}$ disconnects the C-terminal degron from the fusion, and thus prevents degradation and consequently confers resistance.

Such a system can be used to evolve a PCS to either resist cleavage, to become an efficient cleavage substrate for a given protease, or alternatively to evolve a protease that cleaves one PCS but not another. To get this system to work, however, multiple aspects had to be optimized.

First, we had to find a resistance protein that tolerates fusions on both termini and allows the stringency of selection to be tuned by changing the antibiotics concentration. Out of several resistance proteins tested, the hygromycin-B 4O-kinase (HygB-kinase; Rao et al., 1983) and the zeocin/bleomycin-binding protein (Gatignol et al., 1988) turned out to be best suited. In the following we used HygB-kinase as a marker. One reason was substantially lower costs for hygromycin B as compared with zeocin.

Second, the intracellular protease concentration during selection turned out to be a critical parameter. Since selection for cleavage is most stringent at low protease concentrations while selection against cleavage is most stringent at high protease levels, we expressed the protease under control of an IPTG-inducible promoter. This allows adjusting and broadening the stringency of selection. For protease evolution experiments, we started the selection in the absence of IPTG (i.e., with just leaky expression) and then continued selection in the presence of IPTG. However, standard expression vector produced too much

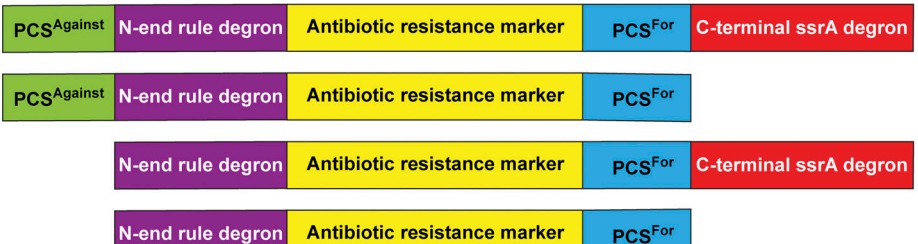

## A  Protease-specificity sensor for co-expressed proteases

| PCS^Against | N-end rule degron | Antibiotic resistance marker | PCS^For | C-terminal ssrA degron |

| PCS^Against | N-end rule degron | Antibiotic resistance marker | PCS^For |

| N-end rule degron | Antibiotic resistance marker | PCS^For | C-terminal ssrA degron |

| N-end rule degron | Antibiotic resistance marker | PCS^For |

### Stability of the sensor

Uncleaved sensor:
**Degraded** through C-terminal degron

PCS^For-cleaved sensor:
**Stable.** Confers antibiotic resistance

PCS^Against-cleaved sensor:
**Degraded** through both degrons

PCS^Against- and PCS^For-cleaved sensor:
**Degraded** through N-terminal degron

## B

| Number | PCS^Against | N-Degron | Marker | PCS^For | C-Degron | Protease | E. coli grown on hygB plates |
|--------|-------------|----------|--------|---------|----------|----------|------------------------------|
| 1 | bdSUMO wild type | None | HygB-kinase | bdSUMO G96A ; G97A | None | bdSENP1 | |
| 2 | bdSUMO G96A ; G97A | FLFVQ | HygB-kinase | bdSUMO wild type | ssrA signal | bdSENP1 | |
| 3 | bdSUMO wild type | FLFVQ | HygB-kinase | bdSUMO wild type | ssrA signal | bdSENP1 | |
| 4 | bdSUMO wild type | FLFVQ | HygB-kinase | bdSUMO G96A ; G97A | ssrA signal | bdSENP1 | |
| 5 | bdSUMO G96A ; G97A | FLFVQ | HygB-kinase | bdSUMO G96A ; G97A | ssrA signal | bdSENP1 | |

$10^{-2}$  $10^{-3}$ $10^{-4}$ $10^{-5}$
Dilution

Figure 1. **A protease-specificity sensor for evolving proteases and cleavage sites.** The sensor couples the stability of an antibiotic-resistance protein (and thus bacterial survival in the presence of antibiotics) to cleavage at two PCSs. **(A)** Order and functions of modules in the sensor fusion protein. Cleavage of the sensor by the coexpressed protease can lead to three different products. Only one of them is stable and able to confer antibiotic resistance. **(B)** Validation of the sensor concept. Five sensor plasmids were constructed and cotransformed with a second plasmid expressing the bdSENP1 protease under IPTG control. The hygromycin-B 4O-kinase (HygB-kinase) confers hygromycin B resistance, provided it is stable. Either WT bdSUMO (an efficient bdSENP1 substrate) or the noncleavable G96A G97A mutant was used as PCSs. The FLFQV peptide served as an N-terminal degron (N-end-rule degradation peptide); it initiates degradation only if the preceding PCS^Against is cleaved and the peptide becomes the extreme N-terminus of the remaining fusion. The C-terminal degron (ssrA degradation signal) is active until cleavage of PCS^For disconnects it from the fusion and saves the HygB-kinase from degradation. Transformed cells were spotted in serial dilutions on plates containing 600 µg/ml hygromycin B and 100 µM IPTG. Cells grew only (1) if no degron was present, or (2) if the C-terminal PCS^For was cleavable and the N-terminal PCS^Against was protease resistant.

protease for differentiating between low- and high-activity variants. We therefore optimized the ribosome-binding site and obtained optimal results with a weaker one of ~50-fold reduced translation efficiency.

The N-terminal degron also required optimization. We tested several destabilizing residues (F, L, W, Y, R, or K) at the $P_1'$ position following PCS^Against, but in all cases, the cleaved fusion protein was degraded too slowly for hygromycin sensitivity to be observed. The solution was an enhanced degron (FLFVQ), where further hydrophobic residues follow an N-terminal phenylalanine (Wang et al., 2008).

Finally, we had to solve an issue with the C-terminal degron, where flexible linkers in front of the ssrA degradation signal were apparently cleaved by endogenous proteases from *E. coli*, leading to a background of hygromycin B resistance without a protease plasmid. In the case of bdSUMO (the SUMO protein from *Brachypodium distachyon*), the solution was to delete the disordered acidic region (residues 1–19) that precedes the ubiquitin fold. The fully optimized system was then validated extensively using several controls as shown in Fig. 1 B.

### SUMO^Eu variants that resist cleavage by animal and yeast SUMO proteases

In a first set of evolution experiments, we evolved bdSUMO and its corresponding SUMO-protease (bdSENP1) to orthogonality to the yeast and human SUMO systems. We chose bdSUMO as a starting point because its fusions are cleaved by scUlp1 (the major SUMO-deconjugating enzyme in yeast) already ≈10 times less efficiently than *S. cerevisiae* (scSUMO) fusions (Frey and Görlich, 2014a), suggesting that resistance could be achieved with fewer mutations. We randomized three (predicted) protease-contacting residues of bdSUMO (T60, D67, and Q75) based on already crystallized SUMO–SUMO protease complexes (see Fig. S1 A), cloned the resulting library as PCS^Against into the sensor plasmid, and selected against cleavage by the SUMOstar protease the so far most promiscuous SUMO-cleaving enzyme (Fig. 2 D and Fig. 3 B; Peroutka et al., 2008). Sequence analysis of highly hygromycin B–resistant clones revealed a strong selection for a D67K exchange, while substitutions at T60 and Q75 were more variable (Fig. 2 A). Phage display selection for scUlp1- and hsSENP2-resistant bdSUMO variants revealed the same strong D67K preference (Fig. 2, B and C).

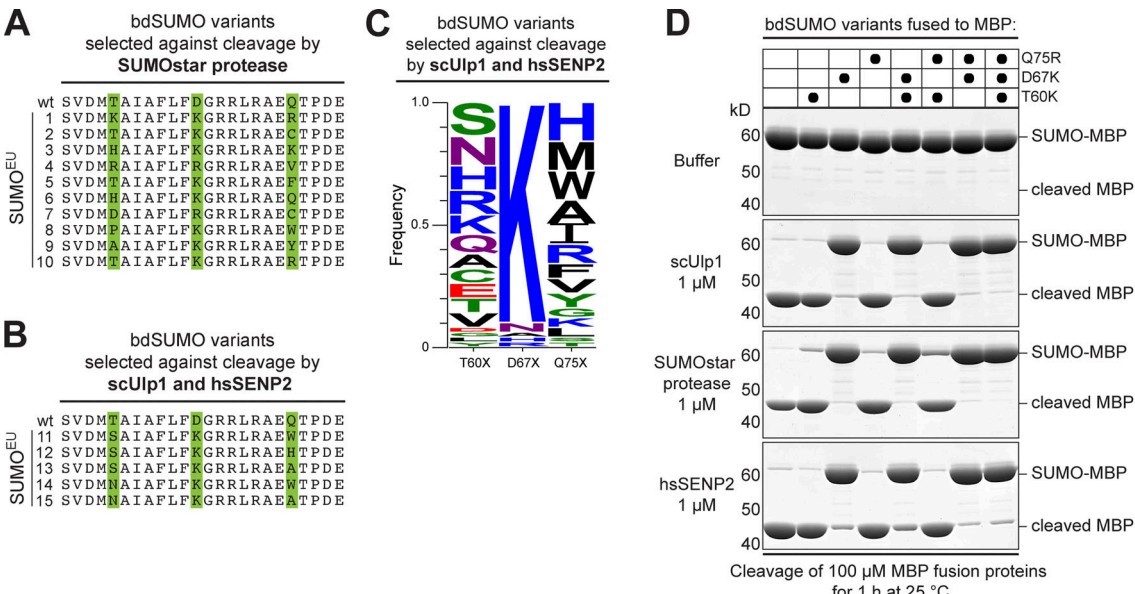

**Figure 2. Cleavage-resistant SUMO[Eu] mutants. (A)** A bdSUMO library with randomized T60, D67, and Q75 positions (marked in green) was selected against cleavage by the SUMOstar protease. Panel shows WT bdSUMO (residues 56–79) aligned with selected SUMO[Eu] mutants. A strong bias for a D67K exchange is evident. **(B)** SUMO[Eu] mutants selected against cleavage by scUlp1 and hsSENP2. Randomized residues are as described in panel A. **(C)** Selection was the same as in panel B, but 96 clones were analyzed, and exchanges are represented by WebLogo (Crooks et al., 2004). **(D)** 100 µM of indicated SUMO–MBP fusions were incubated either in buffer or with 1 µM of the indicated proteases. Analysis was by SDS-PAGE/Coomassie staining with 2 µg sample being loaded. The D67K exchange was sufficient to impede cleavage by the catalytic domains of scUlp1 (residues 403–621), hsSENP2 (residues 361–589), or SUMOstar protease.

In vitro cleavage assays confirmed that the D67K exchange confers cleavage resistance not only toward the SUMOstar protease but also toward WT scUlp1 and hsSENP2. The obtained level of resistance was remarkable, i.e., substrate cleavage remained negligible even when 1 µM of the proteases acted for 1 h at 25°C (Fig. 2 D), which is at least 50 times more protease than needed for complete cleavage of the cognate substrate (Frey and Görlich, 2014a). In the following, we will refer to these cleavage-resistant mutants as SUMO[Eu] variants to indicate their stability in a eukaryotic cytosol.

### Evolution of SENP[Eu] proteases that rapidly cleave SUMO[Eu] variants

For the subsequent protease evolution steps, we diversified bdSENP1 at four (predicted) SUMO-interacting positions (R269, N280, R346 and K350; see Fig. S1 B). The resulting SENP[Eu] library was transformed into *E. coli*, together with a protease-specificity sensor containing SUMO[Eu1] (bdSUMO T60K, D67K, and Q75R) as the PCS[For] cleavage target. Counterselection was against cleavage of scSUMO (PCS[Against] module).

We selected for hygromycin B resistance initially in liquid medium (both in absence and presence of 100 µM IPTG), reamplified the protease-encoding sequences, repeated the selection, and finally plated surviving bacteria onto hygromycin B/IPTG agar plates. Sequencing of ~100 clones eventually identified 10 protease SENP[Eu] variants. Six variants (B, G, H, i, J, and K) were isolated more than once.

As expected, only a very small fraction of the 160,000 possible residue combinations appears to be effective. All selected mutants are shifted to a more negative charge (mean: –3, range:

–2 to –4), which restores charge complementarity to the SUMO[Eu1] mutant that has a +3 charge shift as compared with the WT. Otherwise, however, the observed mutations did not converge to a consensus (Fig. 3 A, lower panel). Instead, distinct sets of mutations (including also accidental single-residue deletions in loops next to the randomized positions) appear to achieve a similar change in substrate specificity of the protease. This can be rationalized by considering that the rather large protease–substrate interface is far from an affinity optimum because rapid substrate turnover requires not only fast binding but also high off-rates. While the affinity optimum can probably be represented by just a single set of interaction residues, there are obviously several ways to deviate from a too strong binding.

### Properties of SENP[Eu] proteases

For a detailed assessment of specificity and activity, we tested the SENP[Eu] variants against a range of SUMO proteins fused to MBP (Fig. 3 A). The data illustrate that all SUMO[Eu] variants are highly resistant against WT scUlp1, the SUMOstar protease, and hsSENP2 even when a protease concentration of 1 µM was used (Fig. 3 B), which is 50 times higher than needed for complete cleavage of the cognate WT SUMO protein (Frey and Görlich, 2014a).

In contrast, we observed a fast cleavage of the SUMO[Eu1] fusion by five of the six selected SENP[Eu] proteases (Fig. 3 B), even with the limiting protease concentration of 20 nM that was used to discern differences in cleavage efficacies. SENP[EuH] showed the highest activity and reached an ~4,000-fold substrate turnover within 1 h. This corresponds to a ≥100-fold higher activity than TEV-protease. The T67X and Q75X exchanges in

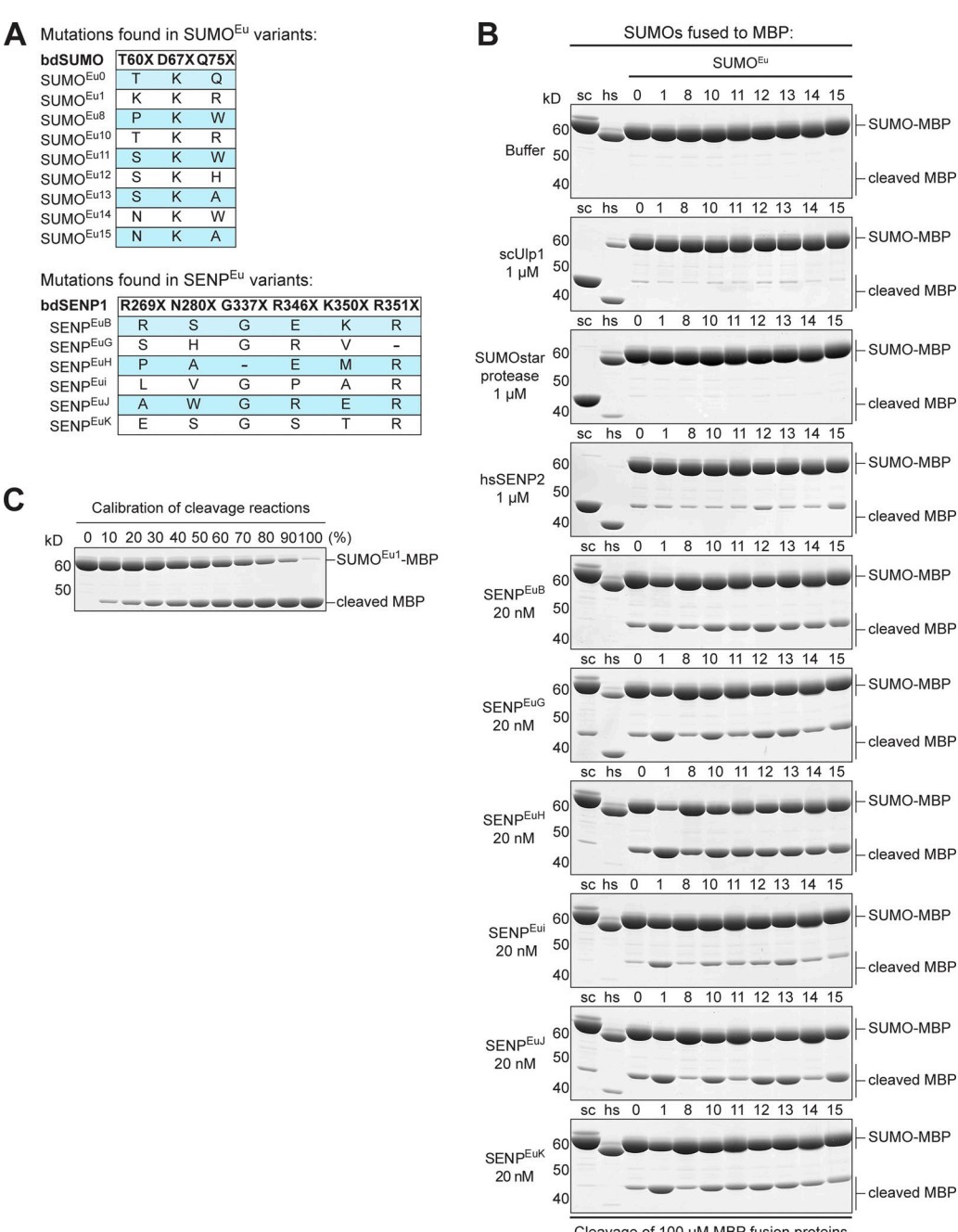

Figure 3. **Characterization of SUMO[Eu] and SENP[Eu] protease mutants. (A)** Mutations present in the SUMO[Eu] and SENP[Eu] variants that were tested in panel B. The dash denotes an amino acid deletion. **(B)** 100 µM of selected SUMO[Eu] mutant–MBP fusions were incubated with indicated SENP[Eu] protease variants, used at a limiting concentration (20 nM) to discern differences in activity. The most active proteases were SENP[EuH] and SENP[EuB] in combination with the SUMO[Eu1] substrate. The SUMO[Eu] mutants were, however, highly resistant even against a rather high concentration (1 µM) of scUlp1, SUMOstar protease, or hsSENP2. Conversely, WT *S. cerevisiae* SUMO (sc) and human SUMO1 (hs) controls were hardly cleaved by either SENP[EuH] or SENP[EuB]. Analysis was as in Fig. 2 C. **(C)** For calibration, undigested and fully digested SUMO[Eu1]–MBP fusion were mixed at indicated ratios. For unknown reasons, there is always a small fraction (1–2%) of uncleavable substrate.

bdSUMO clearly affect substrate recognition. SENP[Eu] B, G, H, and i, for example, cleave SUMO[Eu1] (D67K, T60K, and Q75R) faster than the plain bdSUMO D67K variant (Fig. 3 B).

Several other combinations of SUMO[Eu] variants 1 or 10–15 with SENP[Eu] variants B, H, G, J, or K show also a very rapid substrate turnover (Fig. 3 B). These alternatives give researchers

a choice in downstream applications, for example, when the charge of the SUMO fusion partner matters.

The SENP[Eu] B, H, and K variants cut WT scSUMO fusions ~1,000 times less efficiently than their preferred substrate (Fig. 3 B). This agrees with the applied evolution scheme and holds even in an in vivo context: SENP[EuB] can be

overexpressed in yeast from a strong Gal4 promotor, without causing the lethal de-sumoylation observed upon scUlp1, SUMOstar, or bdSENP1 overexpression (Fig. S2 A). This opens interesting experimental avenues, such as a broader re-engineering of the SUMO system in living cells, conditionally subjecting (SUMO$^{Eu}$-tagged) proteins to degradation by the N-end–rule pathway, or detaching them from an anchor site or transport signal. So far, TEV-protease has been the first choice for the latter applications (see, e.g., Taxis and Knop, 2012). With the far more active SENP$^{EuB}$ protease, however, we expect faster and more complete responses.

## SUMO$^{Eu1}$ is a highly stable but cleavable tag in eukaryotic hosts

We then expressed various SUMO–YFP fusions in yeast and found that WT scSUMO or bdSUMO fusions were cleaved completely (Fig. S2 B). The SUMOstar fusion was stabilized, but still, around 30% cleavage was evident. With less than 10% cleavage, the SUMO$^{Eu1}$ fusion turned out as the most stable construct. This indicates that SUMO$^{Eu1}$ is resistant not only toward scUlp1 but also toward the paralogous scUlp2 protease that was not included in the selection scheme.

Subsequently, we tested a double-tag purification scheme for a model protein complex, exploiting the orthogonality between the SUMOstar and the SUMO$^{Eu}$ systems (Fig. S2 C). To this end, we coexpressed a ZZ–SUMOstar–YFP fusion with a His–SUMO$^{Eu1}$-tagged anti-YFP nanobody in yeast, bound the complex to Ni$^{+2}$ chelate beads, and eluted it by His-tag cleavage with SENP$^{EuB}$. This first eluate contained the expected complex, but also an excess of fused and free YFP as well as some minor contaminations. This eluate was then bound to an anti–protein A (ZZ/ED) matrix (based on the immobilized ZpA963 affibody; Lindborg et al., 2013), and a pure and stoichiometric YFP-nanobody complex was finally eluted by cleaving the ZZ–SUMOstar tag with SUMOstar protease.

To assess if the SUMO$^{Eu1}$ tag is compatible also with other eukaryotic systems, we tested the behavior of various SUMO–MBP fusions in extracts prepared from wheat germ, *Xenopus laevis* eggs, human HeLa, and *Drosophila melanogaster* S2 cells (Fig. S3 A). All these extracts cleaved WT–SUMO fusions, but kept SUMO$^{Eu1}$ fusions entirely stable. This suggests that SUMO$^{Eu}$-based expression systems should be feasible not only in yeast but also in animal cells.

As a test case, we used transient transfection in human HEK-293T cells to express a fusion of a His–EGFP–SUMO$^{Eu1}$ module and the nonpolymerizable human "AP-actin" mutant (Joel et al., 2004; Fig. S3 B). A lysate was prepared and bound to Ni(II) chelate beads. The imidazole eluate contained the expected fusion protein, but also a heavy background of apparently histidine-rich endogenous proteins. Elution with 200 nM SENP$^{EuB}$, however, released the actin mutant selectively, while nearly all the background remained matrix-bound and became desorbed only by the subsequent imidazole post-elution step. Thus, the SUMO$^{Eu}$ system provides an easy way of purifying a protein expressed in human cells within less than 2 h, and without using expensive matrices such as immobilized anti-FLAG antibodies.

## Pdr6 is an importin for the SUMO E2 ligase Ubc9

The pleiotropic drug resistance protein 6 (Pdr6/Kap122) was originally identified through its genetic interactions with Pdr1 (Chen et al., 1991) and subsequently recognized as a HEAT-repeat protein of the importin β superfamily (Görlich et al., 1997). Later studies demonstrated Pdr6-mediated nuclear import of the Toa1/Toa2 transcription factor dimer (Titov and Blobel, 1999) as well as of Wtm1 in complex with the RNR2 and RNR4 subunits of the ribonucleotide reductase (Zhang et al., 2006). We suspected, however, that Pdr6 carries a wider range of substrates and therefore adapted a SUMO$^{Eu}$-assisted affinity chromatography for revisiting the cargo spectrum of Pdr6 (Fig. 4).

To this end, three modules were fused in tandem to the N-terminus of Pdr6, namely, the ED domains from *Staphylococcus aureus* protein A, the SUMO$^{Eu1}$ cleavage module, and a His$_{12}$ tag. The fusion was then incubated with a yeast lysate (containing potential cargoes). In such extract, RanGAP will dominate and create a cytoplasmic (low RanGTP) environment that favors binding of import substrates. To mimic also a nuclear environment, one sample was supplemented with His-tagged RanGTP (added as a GTPase-locked mutant).

Formed cargo–Pdr6 complexes were retrieved with the anti–protein A (ZZ/ED) matrix and subsequently eluted by on-column cleavage with SENP$^{EuB}$ (first chromatographic step in Fig. 4). His-tags remained on Pdr6 and Ran, and these were used to rebind the complexes to a Ni(II) chelate matrix. After washing, 3 M guanidinium–HCl (Gdn-HCl) was applied, which keeps the His-tag Ni(II) interaction intact, but releases prey from the bait (second chromatographic step in Fig. 4). This strategy has the advantages that (1) a mass spectrometric protein identification of potential cargoes can directly be performed on the guanidinium eluates (after appropriate dilution) and that (2) the intense bands of the baits do not obscure fainter interaction partners.

This way and by mass spectrometry, we confirmed Toa1 and Wtm1 as Pdr6 import substrates (Fig. 4). In addition, we identified the SUMO E2-ligase Ubc9 as a RanGTP-sensitive binder and thus as a potential import cargo. Binding assays with only recombinant components and immobilized Ubc9 confirmed the Ubc9–Pdr6 interaction as being direct (Fig. 5 A). Moreover, Ubc9 is not recognized by any other so far described yeast importins (Lph2, Mtr10, Msn5, Nmd5, Yrb4, Kap114, Smx1, Pse1, importin β, or the importin α–β complex; Fig. 5 B). This argues against redundancy in Ubc9 import and can be seen as a further specificity control. In the accompanying manuscript, we characterized the Ubc9–Pdr6 complex further and report its crystal structure (Aksu et al., 2019).

An importin has to drag its cargo initially into the FG domain–based permeability barrier of NPCs. Pdr6 indeed exhibits such an activity: while GFP-fused Ubc9 remained rather excluded from an in vitro–assembled Nup116 FG phase (partition coefficient of ∼0.7), Pdr6 boosted the Ubc9 partitioning to a coefficient of ∼40 (Fig. 5 C).

In a next step, we tagged a chromosomal copy of Ubc9 with GFP in yeast cells and visualized its localization by confocal laser-scanning microscopy (CLSM), using tCherry–NES fusion

Figure 4. **Identification of novel transport substrates for Pdr6/Kap122.** ED-SUMO$^{Eu1}$-His$_{12}$–tagged Pdr6 was incubated as a bait with an extract from yeast cells to recruit either import cargoes (without further addition) or export cargoes (+4 µM RanGTP). Formed complexes were retrieved by an anti ZZ/ED-tag affibody matrix and eluted by cleaving the ED-tag with SENP$^{EuB}$. Eluted complexes were then recaptured via the remaining His-tags. Potential cargoes were released by 3 M guanidinium–HCl (Gdn-HCl), while His-tagged Pdr6 and His-tagged Ran remained bound to the Ni$^{2+}$-matrix and were subsequently post-eluted with imidazole. Analysis was by SDS-PAGE/Coomassie staining. Import cargoes and export cargoes were identified from excised bands by mass spectrometry.

as a nuclear exclusion marker and thus as a reference (Fig. 5 D). This confirmed the previously reported predominantly nuclear localization of Ubc9 (Seufert et al., 1995). In Pdr6-knockout (pdr6Δ) yeast cells, however, the nuclear signal was decreased while the cytoplasmic signal appeared brighter. This confirms that Pdr6 indeed imports Ubc9 into nuclei.

### Pdr6 appears to be an exportin for the eisosome-constituents Pil1 and Lsp1

The interaction assay of Fig. 4 revealed also several RanGTP-dependent Pdr6-binders and thus potential nuclear export substrates, which include Pil1, Lsp1, eIF5A, and eEF2. This was remarkable because so far only import substrates had been reported for Pdr6.

Pil1 and Lsp1 are lipid-binding BAR domain proteins; they are paralogous to each other, form homo- and heterodimers, and are constituents of so-called eisosomes that mark endocytic sites at the plasma membrane (Zhang et al., 2004; Moreira et al., 2009; Ziółkowska et al., 2011). We expressed them individually in E. coli, immobilized them, and observed that each of them recruited Pdr6 in a strongly RanGTP-stimulated manner (Fig. S4, A

and B). This points to direct Pdr6–Pil1 and Pdr6–Lsp1 interactions, and it further suggests that Pdr6 binds these two proteins inside nuclei and transfers them to the cytoplasm. Additionally, such interactions are highly specific as Pdr6 was the only yeast exportin able to recognize Lsp1 and Pil1 in the presence of RanGTP (Fig. S4 C).

### Pdr6 is an exportin for eIF5A and eEF2

eIF5A is a universally conserved protein, also called hypusine-containing protein Hyp2 in yeast or EF-P in eubacteria (Glick and Ganoza, 1975; Kemper et al., 1976; Smit-McBride et al., 1989). Already early on, eIF5A was described as a translation factor, but only more recently, it was shown to be required for translating proline-rich protein stretches (Doerfel et al., 2013; Gutierrez et al., 2013; Ude et al., 2013).

eIF5A was, in the presence of RanGTP, by far the most prominent Pdr6 binder (Fig. 4). The Pdr6–eIF5A interaction is reproducible with recombinant components and thus direct (Fig. 6 A). The assay also shows that eIF5A is recognized only by Pdr6, but not by any of the previously characterized yeast exportins (Fig. 6 B).

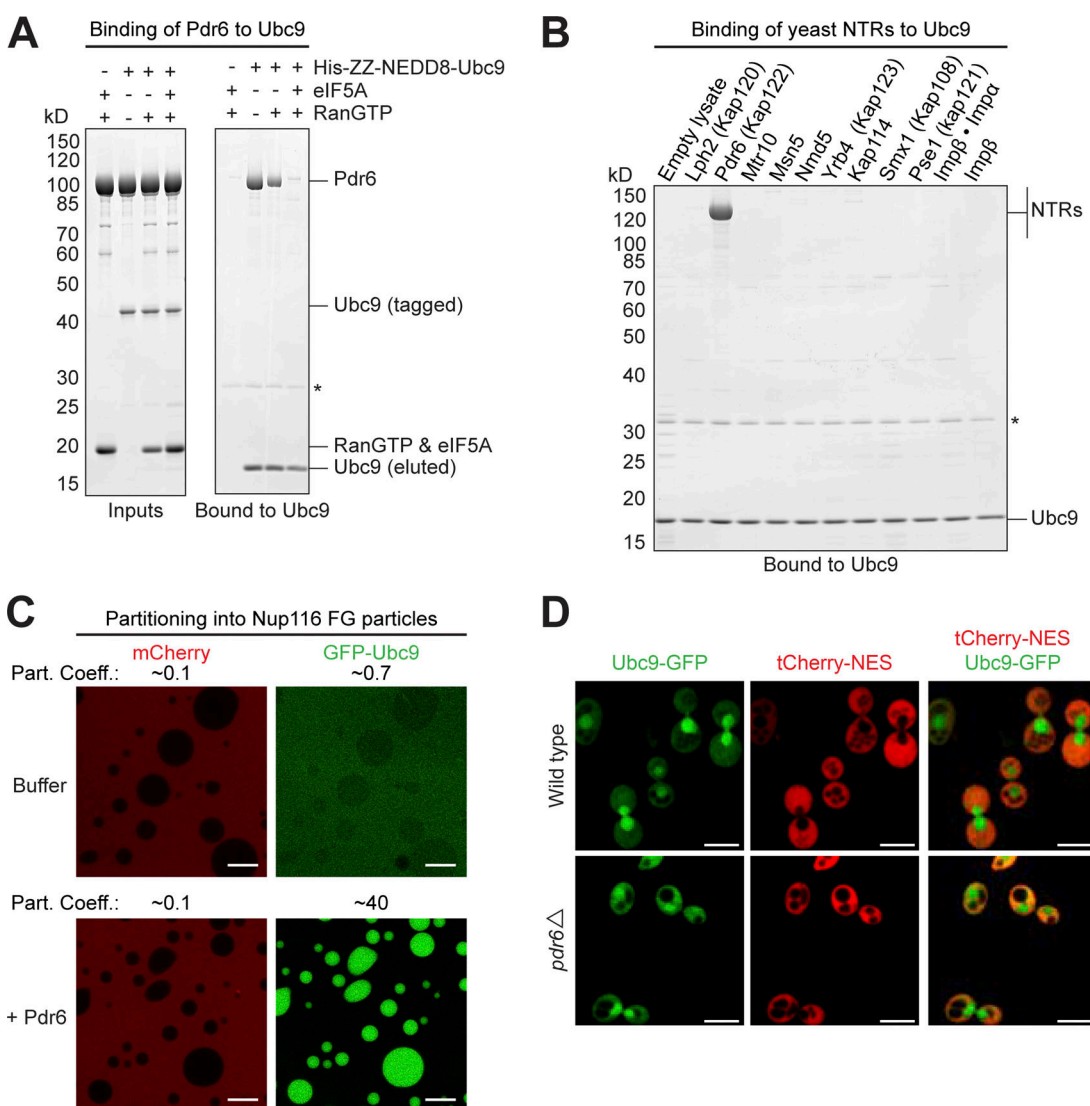

**Figure 5. Ubc9 is a specific import cargo for Pdr6. (A)** 3 µM recombinant Pdr6 was mixed with indicated combinations of 3 µM RanGTP, 3 µM eIF5A, and 1.5 µM $H_{14}$-ZZ-NEDD8–tagged Ubc9 (inputs). Formed complexes were retrieved with an anti-ZZ/ED affibody matrix and eluted by NEDP1-mediated tag-cleavage. Analysis was by SDS-PAGE/Coomassie staining. Relevant bands are labeled. The asterisk represents the protease used for elution. RanGTP and eIF5A comigrate on the gel. Pdr6 bound Ubc9 specifically. The interaction was impeded by RanGTP and more strongly by the combination of RanGTP and the export cargo eIF5A. **(B)** $H_{14}$-ZZ-NEDD8–tagged Ubc9 was added to *E. coli* lysates containing each a different yeast NTR. Formed complexes were isolated and analyzed as in panel A. Note that Pdr6, but no other importin, got recruited to Ubc9. **(C)** Ubc9 was fused to GFP (efGFP_8Q variant; Frey et al., 2018). 1.2 µM GFP–Ubc9 fusion was mixed with 3 µM mCherry and incubated with Nup116 FG particles that recapitulate the permeability barrier of NPCs (Schmidt and Görlich, 2015). Without further addition, both mobile species remained excluded. With 3 µM Pdr6, however, GFP–Ubc9 accumulated inside the FG particles. Partition coefficients (Part. Coeff.) of mCherry and GFP-Ubc9 are given. Analysis was by CLSM. Bar, 10 µm. **(D)** Ubc9 was genomically tagged with eGFP and located in living yeast cells by CLSM. A tetrameric tCherry–NES fusion served as a cytoplasmic marker. The merged images revealed a predominantly nuclear localization of Ubc9 in WT and a redistribution to the cytoplasm in Pdr6-knockout cells (*pdr6Δ*). Bar, 5 µm.

In a next step, we tagged one of two chromosomal copies of eIF5A with a C-terminal GFP, and analyzed the resulting yeast strain by CLSM (Fig. 6 C). This revealed a strong cytoplasmic signal and clear exclusion from the nuclear compartment. Strikingly, this nuclear exclusion was lost in a *pdr6Δ* strain, suggesting that eIF5A can efficiently enter nuclei and that Pdr6 mediates its retrieval to the cytoplasm. This can be seen as a formal proof for an exportin function of Pdr6, and places Pdr6 in the category of a bidirectional transporter. Furthermore, it suggests that yeast Pdr6 combines the function of two mammalian NTRs, namely of importin 13, which imports Ubc9 (Mingot et al., 2001), and of exportin 4 (Xpo4), which exports eIF5A (Lipowsky et al., 2000; Aksu et al., 2016).

The translation elongation factor eEF2 (Skogerson and Moldave, 1968) is yet another RanGTP-dependent Pdr6 binder (Fig. 4). On SDS gels, it migrates close to Pdr6; in a traditional pull-down assay, it would have been obscured by the Pdr6 band. CLSM shows the eEF2–GFP fusion to be well excluded from nuclei (Fig. 6 D). This exclusion is lost in a *pdr6Δ* strain, suggesting that Pdr6 also functions as an exportin for eEF2.

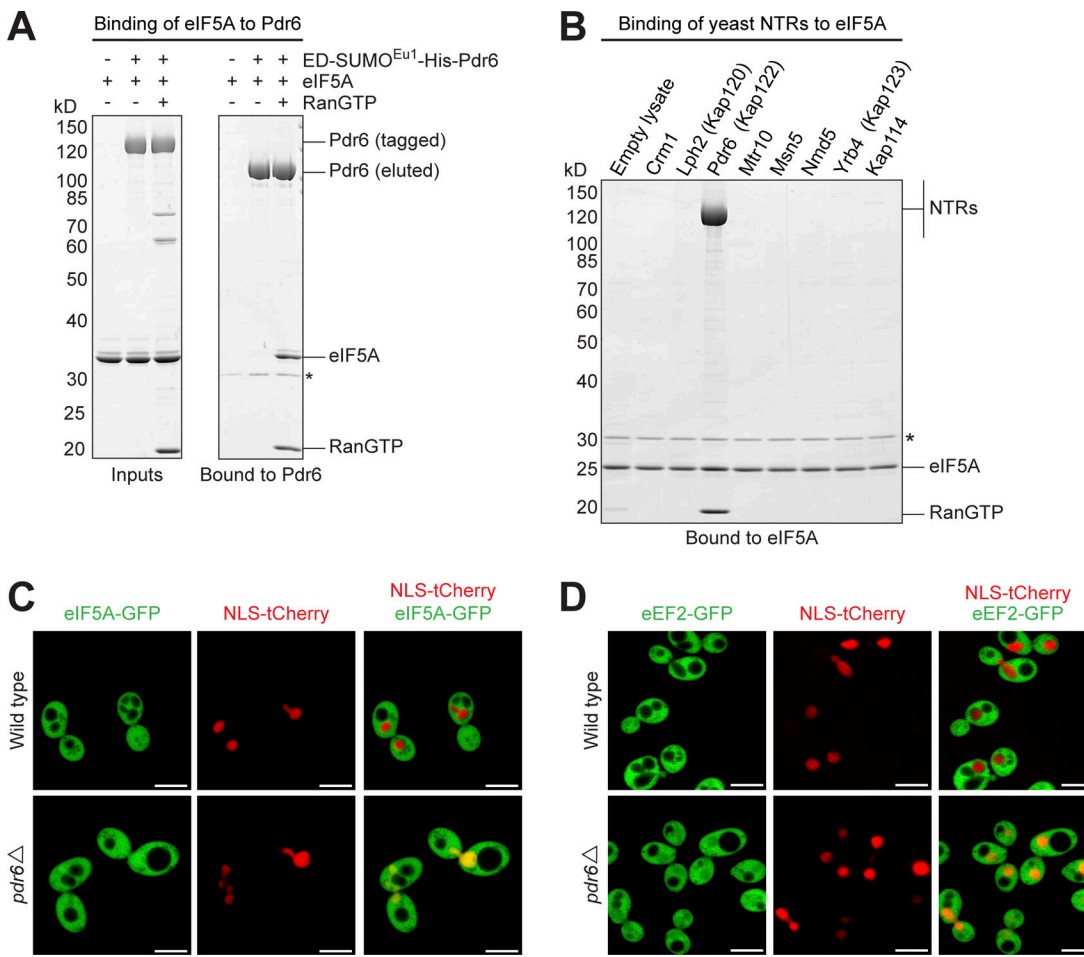

Figure 6. **Pdr6 mediates nuclear export of eIF5A and eEF2. (A)** ED-SUMO$^{Eu1}$-His$_{12}$–tagged Pdr6 was mixed with His$_{14}$-bdNEDD8-eIF5A and RanGTP as indicated (Inputs). Formed complexes were retrieved by anti-ZZ/ED affibody beads and eluted by SENP$^{EuB}$ (indicated by an asterisk). Pdr6 bound eIF5A in a RanGTP-dependent and thus exportin-like manner. **(B)** H$_{14}$-ZZ-NEDD8–tagged eIF5A was added to *E. coli* lysates containing RanGTP and a different yeast NTR each. Formed complexes were isolated and analyzed as above. Note that Pdr6, but no other exportin, got recruited to eIF5A. **(C)** CLSM of living yeast cells with genomically eGFP-tagged eIF5A. A tCherry–NLS fusion served as a nuclear marker. eIF5A showed a bright cytoplasmic signal and nuclear exclusion in WT cells. Nuclear exclusion was lost in the *pdr6Δ* strain. **(D)** Analysis of eEF2 localization. The eEF2–GFP fusion is exclusively cytoplasmic in WT yeast cells, but shows a clear nuclear signal in the absence of Pdr6. Bars, 5 µm.

## Discussion

### The SUMO$^{Eu}$ system and possible applications of the selection system

SUMO-fused tags combined with SENP1 proteases are powerful tools for recombinant protein production. They allow to cleanly remove tags following expression and purification or even to use such protease as an eluent from an affinity matrix. The latter strategy not only streamlines workflows but also yields purer products, because the specificities of protease and affinity matrix combine. The use of WT–SUMO fusions is, however, limited to prokaryotic expression, because endogenous proteases of a eukaryotic host cleave them prematurely.

To address such a cross-reaction issue, we conceived an in vivo selection system for engineering PCSs and PCS-cleaving proteases to novel specificities. We used the system to evolve SUMO$^{Eu}$/SENP$^{Eu}$ protease pairs that are orthogonal to WT SUMO systems (Fig. 2 D and Fig. 3 B). Accordingly, SUMO$^{Eu}$ fusions are stable when expressed in yeast or human cells but get rapidly cleaved by the new SENP$^{Eu}$ proteases. The evolution

strategy can now be applied to novel problems, for example, for adapting other tag-cleaving proteases (NEDP1, Atg4, ubiquitin proteases) and their cleavage sites to eukaryotic expression systems, for studying viral proteases, or for evolving proteases that cleave a chosen cellular target.

We routinely obtain 25–50 mg SENP$^{Eu}$ proteases from 1 liter of *E. coli* culture (with standard medium and shaking flasks)—enough to cleave tags from recombinant proteins at a scale of tens to hundreds of grams. These large amounts indicate that the in vivo evolution system selected not only for the desired protease specificities but also for optimal protein expression, stability, and folding. One-step purifications (through Ni-chelate chromatography) gave already highly pure proteases; in contrast to the SUMOstar protease, no copurification of nucleic acids or other bacterial contaminants were evident.

We validated the SUMO$^{Eu}$ system in several practical applications, first in recombinant protein expression and purification in the yeast *S. cerevisiae* as well as in human HEK-293T cells (Fig. S2 C and Fig. S3 B). Given that SUMO$^{Eu1}$ fusions stayed stable in

*Drosophila* S2 extracts (Fig. S3 A), we would assume that the SUMO[Eu] system is applicable also for insect cell expression.

### Ubc9, a novel import substrate for Pdr6

In a final application, we used SUMO[Eu]/SENP[Eu]-assisted affinity chromatography to re-explore the transport cargo spectrum of Pdr6/Kap122 (Fig. 4). This identified not only the previously ascertained import substrates (Titov and Blobel, 1999; Zhang et al., 2006) but also Ubc9 as a new import cargo candidate for Pdr6 (Fig. 5 A). This assignment was confirmed by the observation that a deletion of the Pdr6 gene shifts Ubc9 from predominantly nuclear to a more cytoplasmic localization (Fig. 5 C).

The remaining nuclear Ubc9 signal in *pdr6Δ* cells can be explained by the rather small size of Ubc9 (18 kD) and thus by unassisted NPC passage—combined with nuclear retention by Ubc9 interaction partners. This, however, raises the question of why such a small protein is actively imported at all. Ubc9 is a SUMO-conjugating E2 enzyme with numerous nuclear but also some cytoplasmic substrates (Flotho and Melchior, 2013; Hendriks and Vertegaal, 2016). The functioning of the SUMO system thus requires a preset nucleocytoplasmic ratio in Ubc9 activity, and we now assume that such a gradient is best maintained by an active transport process. As discussed in the accompanying paper (Aksu et al., 2019), we regard it as an attractive possibility that Pdr6 also functions as a chaperone to modulate Ubc9 activity in a compartment-specific manner.

### Why do cytoplasmic factors need active retrieval from nuclei?

SUMO[Eu]/SENP[Eu]-assisted affinity chromatography allowed the discovery of RanGTP-dependent Pdr6-binders (i.e., export cargoes) and hence of Pdr6-mediated protein export (Figs. 4, 5, and 6). The four here characterized Pdr6 export cargoes (Pil1/Lsp1, eIF5A, and eEF2) have an exclusively cytoplasmic localization in WT cells, which is consistent with their vital cytoplasmic functions. The nuclear accumulation of eIF5A and eEF2 in *pdr6Δ* cells confirmed Pdr6 as an export receptor. It also indicates a shuttling of eIF5A and eEF2 between nucleus and cytoplasm; however, it does not imply a nuclear function. Instead, we assume that nuclear pools of Pil1/Lsp1, eIF5A, and eEF2 represent mislocalized populations.

There is no indication of an active import of these proteins. Their nuclear entry is therefore perhaps best explained by the NPC barrier being unable to retain these proteins permanently in the cytoplasm (Frey et al., 2018). Active, Pdr6-mediated retrieval to the cytoplasm should, therefore, be seen as a backup system that compensates for this imperfection of the barrier.

eIF5A is a very small protein (15 kD), and this small size provides a straightforward explanation for a fast leakage into nuclei. eEF2, however, is far larger (93 kD for the unfused protein and 120 kD for the here-analyzed GFP fusion). Yet, the rather short time of a yeast cell cycle appears to be sufficient for a nucleocytoplasmic equilibration in *pdr6Δ* cells. We see two explanations for this fast exchange. First, eEF2 has a rather elongated shape (Jørgensen et al., 2003), which might make it easier to traverse a sieve-like barrier with a small mesh size. The second explanation relates to our recent finding that the exposure of hydrophobic residues, arginines, and histidines confers

interactions with the FG repeats of the permeability barrier and consequently facilitates barrier passage (Frey et al., 2018). Indeed, the eEF2 structure (Jørgensen et al., 2003) shows numerous exposed residues with such "translocation-friendly" side chains that should accelerate leakage into nuclei. The same shape and surface considerations actually apply also to the ~80-kD Pil1/Lsp1 dimers (Ziółkowska et al., 2011).

Why do cells retrieve these proteins back to the cytoplasm? One reason is that the mislocalized molecules represent a nonfunctional pool and hence an investment of cellular resources without a revenue. Mislocalization is thus a metabolic burden. An additional reason might be that the mislocalized pools exert harmful effects inside nuclei. Pil1/Lsp1, for example, might bind and reshape the inner nuclear membrane, and compete there with physiological membrane ligands. Nuclear translation of nonspliced premRNAs is another potentially harmful process because ribosomes will run here into introns and produce protein fragments that are not only nonfunctional but perhaps even toxic. Pdr6-mediated eIF5A and eEF2 export can be seen as part of the cell's effort to confine translation to the cytoplasm.

In fact, Pdr6 synergizes here with the major exportin Xpo1/CRM1, which expels at least three individual yeast translation factors (eIF5, eIF5B, and polyA-binding protein) and four translation factor complexes (eIF2, eIF2B, eIF3, and eIF4G) to the cytoplasm (Kirli et al., 2015). The combined action of Pdr6 and CRM1 should, therefore, result in a very robust exclusion of nuclear translation.

CRM1 and Pdr6 differ strikingly in how they recognize their cargoes. CRM1 binds linear motifs within disordered regions, namely, leucine-rich nuclear export signals (NESs; Dong et al., 2009; Monecke et al., 2009). In contrast, Pdr6 recognizes the central BAR domain in the cases of Pil1 and Lsp1 (Fig. S4 C). In the case of eIF5A, it contacts both globular domains (see the accompanying paper by Aksu et al. [2019] for a description of the structure); eEF2 lacks disordered regions that could possibly serve as a linear export signal. Therefore, Pdr6 appears to recognize folded domains. How Pdr6 can adapt to the very different folds of Pil1/Lsp1, eIF5A, and eEF2 is an intriguing but still unresolved question.

### Biportins: An apparently common class of NTRs

Pdr6 is also special because it is neither a pure exportin nor a pure importin, but a bidirectional NTR. We now suggest the term "biportin" to describe such kinds of transporters. Biportins appear more economical than unidirectional transporters because they can transport cargo during nuclear entry as well as during exit and thus carry two cargoes per RanGTPase and transport cycle. This lower energy spending per cargo reduces, however, the power for accumulating cargo against a concentration gradient.

Nuclear RanGTP displaces import cargoes from an import receptor. This antagonism (negative cooperativity) explains the need for strong binary interactions between RanGTP and a typical importin with a $K_d$ of around 1 nM (Bischoff and Görlich, 1997; Hahn and Schlenstedt, 2011). In contrast, cargo-free exportins bind RanGTP only very weakly with a $K_d$ of >1 µM. This leaves room for the extreme positive cooperativity in binding of

export cargo and RanGTP, i.e., for the ~1,000-fold enhancement of cargo binding by RanGTP and vice versa (Kutay et al., 1997; Güttler et al., 2010).

The affinity of Pdr6 for RanGTP is in between a typical importin and a typical exportin ($K_d$ = 230 nM; Hahn and Schlenstedt, 2011). This makes perfect sense, as Ran has to bind in positive cooperativity with the export cargoes and antagonistically with respect to the import substrates. The intermediate affinity has, however, also a functional consequence, namely, that RanGTP cannot fully release the import cargo Ubc9 (Fig. 5 A). A complete release requires either the recruitment of an export cargo such as eIF5A or the transfer of the import cargo to some nuclear binding partner.

The intermediate affinity for RanGTP is shared by Msn5 ($K_d$ = 50 nM), which previously was shown to mediate nuclear export as well as import (Kaffman et al., 1998; Yoshida and Blobel, 2001; Hahn and Schlenstedt, 2011). Interestingly, yeast has three additional importins that bind RanGTP with intermediate strength (Hahn and Schlenstedt, 2011), namely, Lph2 (270 nM), Mtr10 (130 nM), and Nmd5 (16 nM). It is now tempting to hypothesize that these three do function in export as well. First experiments support this assumption. This implies that biportins are not an exception, but a rather common class of NTRs.

## Materials and methods

### In vivo system selection in *E. coli* cells
*E. coli* TOP 10 F′ cells (lacI$^q$ Tn10 (tet$^R$) mcrA Δ(mrr-hsdRMS-mcrBC) φ80lacZΔM15 ΔlacX74 deoR nupG recA1 araD139 Δ(ara-leu)7697 galU galK rpsL(Str$^R$) endA1 λ$^-$) were used for all the experiments. To test the functionality of the selection system, cells were transformed with the corresponding vectors (see Table S1) and streaked onto 2YT plates that were supplemented with 50 µg/ml kanamycin (Kan) and 50 µg/ml spectinomycin (Spec) to select for the plasmids. Single colonies were picked and grown in 2YT/Kan/Spec medium for 16 h at 37°C. An aliquot of each bacterial culture was diluted to an OD$_{600}$ of 1.0 and followed by a 10-fold serial dilution scheme using fresh selective 2YT medium. 5 µl of each dilution was then spotted onto 2YT agar plates supplemented with hygromycin B (600 µg/ml) in the presence and absence of 100 µM IPTG. Plates were incubated for 18 h at 37°C and scanned using an Epson Perfection V700 Photo scanner. For the test, the standard ribosome binding site (AGAGGAGA) was modified (AAACAAGT) to achieve a 50 times less efficient translation of the corresponding protease.

For all protein evolution experiments, the libraries were synthesized as double-stranded DNA fragments with specific codons being randomly mutagenized (GeneScript). Libraries were then amplified by PCR and cloned using the Gibson assembly protocol (Gibson, 2011) into either a protease-specificity sensor vector (bdSUMO mutant library) or an inducible expression vector (bdSENP1 mutant library). Cloned libraries were then transformed into cells containing already a desire protease expression plasmid or a given protease-specificity sensor vector. Selection of bdSUMO variants was performed in selective 2YT medium plates supplemented with Kan, Spec, hygromycin B (600 µg/ml) and IPTG (100 µM) for 18 h at 37°C. Evolution of

bdSENP1 variants was first performed in liquid 2YT medium supplemented only with Kan, Spec, and hygromycin B (600 µg/ml) for 2 h at 37°. Then IPTG (100 µM) was added and the selection continued for another for 18 h at 37°C. An aliquot of the resulting culture was used to reamplified the bdSENP1 variants by PCR and clone them into a protease expression plasmid. Cells were transformed and inoculated for another round of selection as described above. After incubation for 18 h at 37°C, cells were plated onto 2YT medium supplemented with hygromycin B (600 µg/ml) and IPTG (100 µM). Individual bdSUMO and bdSENP1 colonies were picked for sequencing and retested using serial dilutions as described above.

### Bacterial protein expression and purification
All proteins were expressed in *E. coli* NEB express cells (*fhuA2 Δlon ompT gal sulA11 R(mcr-73::miniTn10-TetS)2 Δdcm R(zgb-210:: Tn10-TetS) endA1 D(mcrCthe-mrr)114::IS10* from the corresponding expression plasmids (see Table S1). SUMO–MBP fusions and bdSENP1 variants were expressed as His-tag fusions, whereas transport cargoes as well as Pdr6 were expressed either with a His$_{14}$-ZZ-NEDD8 tag or an ED-SUMO$^{Eu1}$-His$_{12}$-tag. After expression, cells were resuspended in LS buffer (45 mM Tris/HCl, pH 7.5, 25 0 mM NaCl, 4.5 mM MgCl$_2$, 20 mM imidazole/HCl, pH 7.5, and 5 mM DTT) and lysed by sonication on ice. A cleared lysate was produced by ultracentrifugation, and rotated with Ni$^{+2}$ chelate beads. After filling into a column, the matrix was washed with HS buffer (45 mM Tris/HCl, pH 7.5, 500 mM NaCl, 4.5 mM MgCl$_2$, 20 mM imidazole/HCl, pH 7.5, and 5 mM DTT). Proteins were eluted either by imidazole elution (LS buffer supplemented with 400 mM imidazole) or by on-column protein cleavage with appropriate proteases (LS buffer supplemented with 500 nM bdNEDP1 or 50 nM bdSENP1; Frey and Görlich, 2014b). As RanGTP, we used the yeast Ran-orthologue Gsp1 carrying a GTPase-blocking Q71L mutation and a deletion of the last 40 residues (Gsp1 Q71L ΔC). Gsp1 was expressed as His$_{14}$–ZZ–SUMOstar fusion, purified as described above and eluted using 100 nM SUMOstar protease.

### In vitro cleavage reactions
Cleavage reactions of SUMO–MBP fusion proteins by a given protease were always performed using purified components in a total volume of 20 µl. Prior to the reaction, fusion proteins and proteases were diluted with cleavage buffer (45 mM Tris/HCl, pH 7.5, 250 mM NaCl, 2 mM MgCl$_2$, 250 mM sucrose, and 10 mM DTT) to twice the final concentration in the reaction. Next, equal volumes of diluted substrate and proteases were mixed in order to start the cleavage reaction and incubated for a specific time and temperature. The cleavage reactions were stopped after adding 180 µl of SDS sample buffer (3% wt/vol SDS, 125 mM Tris/HCl, pH 6.8, 50 mM DTT, 1 M sucrose, and 0.002% bromophenol blue). Generally, samples corresponding to 2 µg of each SUMO–MBP fusion protein were resolved by SDS-PAGE and stained by Coomassie blue G250.

### Synthesis of the anti-ZZ/ED-affinity matrix
This matrix is based on the ZpA963 affibody (Lindborg et al., 2013), recognizing all of the five domains of *Staphylococcus*

protein A, whereby the E and D domains are bound more tightly than the B domain that traditionally has been used as a Z- or ZZ-tag. Since the affinity is moderate (~50 nM), we use tags comprising two protein A domains (ED or ZZ) as well as two affibodies fused in tandem, including a hydrophilic C-terminal spacer and terminal cysteine (pDG2506; see Table S1) for coupling by maleimide chemistry.

The ZpA963 matrix is a superior alternative to IgG-sepharose with less background and no leakage of the coupled ligand. Its preparation involves four steps: activation of the sepharose by epichlorohydrin to yield epoxy-activated sepharose, reaction of the epoxy groups with ammonia to yield amino-sepharose, reaction with a maleimide–N-hydroxysuccinimide bifunctional cross-linker yielding maleimide-activated sepharose, and finally reaction with the ligand.

1 liter sepharose 4B (GE Healthcare) was equilibrated in pure water, brought to a 50% slurry, and transferred to a 5-liter screw bottle. 0.5 M epichlorohydrin and 0.4 M NaOH were added (under a fume hood). The bottle was closed and the mixture was shaken for 1 h at 20°C. The resulting epoxy-activated sepharose was filtered on a glass funnel and thoroughly washed with water. It was then resuspended in 1 liter of 4 M $NH_4Cl$. 500 ml of 10 M aqueous $NH_3$ was added, and the mixture was shaken overnight at 20°C. The resulting amino-sepharose was washed with water until ammonia became nondetectable, and stored in 20% aqueous ethanol until further use.

For maleimide activation, 50 ml amino-sepharose was brought to a 50% slurry in 100 mM potassium phosphate, pH 7.0; 5 mM 3-(maleimido) propionic acid N-succinimidyl ester (IRIS Biotech) was added from a 10 mM solution in dimethylformamide; and the slurry was shaken for 30 min at 20°C. After washing in 100 mM potassium phosphate, pH 7.0, and resuspension in the same buffer, the reduced (but otherwise thiol-free) ligand was coupled to a concentration of 200 µM (referring to the bead volume). Non-reacted maleimide groups were quenched with cysteine. The matrix was thoroughly washed and stored at 4°C in 20% ethanol until use.

### Protein expression in *S. cerevisiae*

The *S. cerevisiae* strain SFY123 (S288C, MATα, ADE2, H2B-CFP::TRP1, *his3Δ200, leu2Δ0, lys2Δ0,met15 Δ0, ura3Δ0*) was used for the overexpression of SUMO–YFP fusion proteins and the Nb–YFP dimeric complex. For this, cells were transformed with the corresponding 2µ expression plasmids containing a *GAL1* promoter (see Table S1). All SUMO–YFP fusions were expressed with an N-terminal ZZ-SUMOstar tag, whereas the anti-YFP/GFP nanobody "enhancer" (Kirchhofer et al., 2010) was expressed as an N-terminally His14-SUMO^Eu1–tagged protein. Single colonies were grown in CSM-Ura, and protein was expressed for 8 h at 30°C in the presence of 2% (wt/vol) galactose. Only for the expression of the dimeric complex, 300 µg/ml of hygromycin B was supplemented to the CSM-Ura medium for the selection of the anti-GFP/YFP nanobody and the ZZ-SUMOstar-YFP plasmids, respectively.

After expression of the SUMO–YFP fusions, the stability of the SUMO tags was tested. Briefly, yeast lysates were prepared using the TCA/NaOH method, and a sample corresponding to $1 \times 10^6$ cells was analyzed by Western blot using an affinity-purified polyclonal rabbit anti-GFP IgG (0.7 µg/ml). The polyclonal anti-GFP IgG was detected with a goat anti-rabbit secondary antibody coupled to IRDye800CW in a 1:5,000 dilution (926-32211; LI-COR).

For the purification of the expressed Nb–YFP dimeric complex, cells were first resuspended in a lysis buffer (50 mM Tris/HCl, pH 7.5, 150 mM NaCl, 20 mM imidazole, and 5 mM DTT) supplemented with 1× concentrated protease inhibitors (S8830; Sigma-Aldrich). Prior to the purification of the dimeric complex, a cellular lysate was prepared by glass beads vortexing and subjected to ultracentrifugation at 4°C for 90 min at 40,000 rpm. The complex was purified using two consecutive affinity chromatographic steps. First, the complex was immobilized using a $Ni^{+2}$ chelate matrix and eluted using lysis buffer containing SENP1^EuB (250 nM, final concentration) for 1 h at 4°C. Next, the resulting eluate was loaded to an anti-ZZ/ED affibody matrix for 1 h at 4°C. After washing out remaining contaminants, the protein complex was desorbed from the matrix by protease-based elution using the SUMOstar protease (100 nM, final concentration) for 1 h at 4°C. The remaining affinity tags were released from the affinity columns by an imidazole-containing SDS sample buffer.

### Yeast viability after overexpressing SUMO proteases

*S. cerevisiae* strain SFY123 (S288C, MATα, ADE2, H2B-CFP::TRP1, *his3Δ200, leu2Δ0, lys2Δ0,met15 Δ0, ura3Δ0*) was transformed with the corresponding 2µ expression plasmids containing a *GAL1* promoter (see Table S1). Single transformed colonies were inoculated in CSM-Ura medium supplemented with 2% (wt/vol) of glucose and further inoculated for 16 h at 30°C. Cells were then pelleted and resuspended using fresh CSM-Ura medium supplemented with 2% (wt/vol) glucose and 2% (wt/vol) raffinose. Resuspended cells were incubated in SD medium containing only 2% (wt/vol) of raffinose until exponential growth phase was reached ($OD_{600}$ of ≈1.0). Next, cells were sequentially diluted in 10-fold steps, and 5 µl of each dilution was spotted in plates containing either galactose (2.0% wt/vol) or glucose (2.0% wt/vol). Plates were incubated for 72 h at 30°C and finally scanned.

### SUMO tag stability in eukaryotic extracts

The stability of the SUMO–MBP fusion proteins was tested in wheat germ extract, *Xenopus* egg extract, HeLa extract, and *Drosophila* S2 extract. The preparation of the lysates was performed essentially as described (Blow and Laskey, 1986; Crevel and Cotterill, 1991). For a 12.5-µl volume reaction, 1 µM of a given SUMO-tagged MBP was incubated with 10 µl of each lysate for 2 h at 25°C in the presence and absence of a protease mix containing 0.1 µM of different SUMO-proteases (scUlp1, SUMOstar protease, bdSENP1, and SENP1^Eu). The reactions were stopped by adding SDS sample buffer to a final volume of 100 µl followed by analysis by Western blot using mouse anti-MBP primary monoclonal IgG (1:5,000 dilution; M1321; Sigma-Aldrich). The primary antibody was detected by a goat anti-mouse secondary antibody coupled to IRDye800CW in a 1:5,000 dilution

(926-32211; LI-COR). Blotted membranes were scanned using the LI-COR imaging system.

## Purification of actin expressed in HEK-293T cells

A nonpolymerizable human AP-actin mutant (Joel et al., 2004) was cloned into an expression vector containing an N-terminal His$_{14}$-eGFP-SUMO$^{Eu1}$-tag under the control of the constitutive promoter eEF1A (see Table S1). The expression construct was transfected into HEK-293T cells using polyethylenimine following the protocol of Longo et al. (2013). Transfected cells were grown in 175 cm$^2$ flasks in DMEM supplemented with 10% (vol/vol) of FBS, penicillin (100 U/ml), streptomycin (100 µg/ml), and Geneticin (1 mg/ml). After 48 h of incubation, cells were resuspended in lysis buffer (10 mM Hepes/NaOH, pH 7.6, 10 mM KCl, 20 mM imidazole 1.5 mM MgCl$_2$, and 2 mM DTT) supplemented with 1× concentrated protease inhibitors (S8830; Sigma-Aldrich). For the purification of AP-actin, cells were lysed by mild sonication, and a cleared lysate was produced by centrifugation at 50,000 rpm for 90 min at 4°C. The lysate was applied to a Ni$^{+2}$ chelate column over 1 h at 4°C. The column was washed thoroughly with low salt buffer (10 mM Hepes/NaOH, pH 7.6, 100 mM KCl, 20 mM imidazole 1.5 mM MgCl$_2$, and 2 mM DTT) followed by a washing step using high-salt buffer (10 mM Hepes/NaOH, pH 7.6, 1 M KCl, 100 mM imidazole 1.5 mM MgCl$_2$, and 2 mM DTT). AP-actin was eluted using either elution buffer (10 mM Hepes/NaOH, pH 7.6, 100 mM KCl, 400 mM imidazole, 1.5 mM MgCl$_2$, and 2 mM DTT) or a low-salt buffer supplemented with SENP1$^{EuB}$ (250 nM, final concentration).

## Identification of Pdr6 cargoes

The yeast cell extract was prepared in GK75 buffer (20 mM Hepes-KOH, pH 7.9, 1.5 mM MgCl$_2$, 75 mM KCl, 5% glycerol, and 0.5 mM DTT) essentially as described in Gottschalk et al. (1999). First, 1 ml of the yeast lysate was incubated with 0.5 µM ED-SUMO$^{Eu}$-His-Pdr6 in the presence or absence of 3 µM His-tagged Gsp1-GTP (Q71L ΔC-terminus mutant) in a final volume of 1.5 ml. 25 µl of anti-ED-tag affibody beads was incubated for 1 h at 4°C with each sample followed by washing off the unbound material and protein elution by on-column protein cleavage for 1 h at 4°C using 120 µl of GK75 buffer supplemented with 20 mM imidazole and 250 mM SENP1$^{EuB}$. Eluates were then incubated with 50 µl of Ni$^{2+}$ chelate beads for 1 h at 4°C, and subsequent protein elution was performed by mixing the beads with 200 µl of Gdn-HCl elution buffer (3 M guanidinium chloride and 50 mM Tris/HCl, pH 8.0) for 5 min at 25°C. Guanidinium hydrochloride–containing eluates were subjected to protein precipitation using isopropanol (90% vol/vol) before analysis by SDS-PAGE and the subsequent mass-spectrometric protein identification (performed by the bioanalytical mass spectrometry service of the Max Planck Institute for Biophysical Chemistry). Ni$^{2+}$ chelate beads were then washed with 500 µl of GK75 buffer to remove the remaining guanidinium chloride from the nickel beads prior to elution of the His-tagged Pdr6 and the Gsp1-GTP using SDS sample buffer supplemented with 400 mM imidazole. Samples of the input materials and eluates were all analyzed by SDS-PAGE/Coomassie staining. Specific protein bands were excised from the gel to identify the Pdr6 cargoes

using liquid chromatography mass spectrometry. The following number of unique tryptic peptides was identified for each cargo: eIF5A (9), eEF2 (17), Lsp1 (5), Pil1 (17), Ubc9 (3), Wtm1 (12), and Toa1(6).

## In vitro binding assays with recombinant components

If not stated otherwise, binding reactions were performed for 1 h at 4° in 200 µl binding buffer (45 mM Tris/HCl, pH 7.5, 100 mM NaCl, 2 mM MgCl$_2$, and 5 mM DTT) using an ED-tagged bait protein (1.5 µM) and a prey protein (3 µM) in the presence or absence of 3 µM RanGTP (Gsp1 Q71L ΔC-terminus mutant). Samples were then incubated with 25 µl of anti-ZZ/ED affibody matrix and eluted by on-column protein cleavage using 50 µl of binding buffer supplemented with SENP$^{EuB}$ (250 nM) or bdNEDP1 (500 nM) for 1 h at 4°C. Input samples and eluates corresponding to 2 µg of the NTR were analyzed by SDS-PAGE/Coomassie staining. To test the cargo-binding specificity of different yeast NTRs, E. coli lysates containing a specific overexpressed eGFP-tagged NTR were used as prey protein for the binding assay. A bacterial lysate containing 3 µM eGFP-NTR was used for each reaction. The binding assay was then performed as described above.

## Construction of yeast strains and CLSM

WT and Pdr6-knockout cells from the yeast strain BY4741 were obtained from the *Saccharomyces* Genomic Deletion Project (http://www-sequence.stanford.edu/group/yeast_deletion_project/; Shoemaker et al., 1996). Yeast codon-optimized GFP was introduced to yeast cells by homologous recombination as a PCR product using as template plasmid PYM25 (Janke et al., 2004) to tag the C-termini of Ubc9, Pil1, Lsp1, eIF5A, and eEF2 as described by Gietz and Schiestl (2007). Cells were selected on CSM-Ura containing 300 µg/ml of hygromycin B and 250 µg/ml of G418 when using the Pdr6-knockout mutant. To screen for positive clones, genomic DNA was extracted from single colonies and tested by PCR using a specific set of primers that amplifies a PCR product only if GFP was correctly inserted at the C-terminus of the targeted gene.

Positive GFP-tagged cells were transformed with a modified version of the PYM-N11 plasmid (Janke et al., 2004) that codes for a tetrameric red fluorescent protein (Frey et al., 2018) fused to a NES (tCherry-NES) or to a nuclear localization signal (NLS-tCherry). Prior to imaging, cells were grown in CSM-Ura medium until mid-log phase (OD$_{600}$ of ∼0.7). Then, living yeast cells were imaged in fresh CSM-Ura medium at 25°C with a Leica SP5 confocal laser-scanning microscope by sequential scans with excitations at 488 nm (for eGFP) and 565 nm (for Cherry) laser lines and a 63× HCX PI apo lambda blue 1.4 oil objective (Leica).

## Online supplemental material

Fig. S1 shows sequence alignment of different SUMO and SUMO proteases orthologues and depiction of interacting motifs within the SUMO/SUMO protease complex. Fig. S2 shows applications of the SUMO$^{Eu}$ system in eukaryotic cells. Fig. S3 shows expression of SUMO$^{Eu}$ fusions in higher eukaryotes. Fig. S4 describes Pil1 and Lsp1 as export cargoes for Pdr6. Fig. S5 shows

localization of eIF5A and eEF2 in living yeast cells. tCherry-NES was used as a cytoplasmic marker. Table S1 lists the constructs used in this study.

## Acknowledgments

We thank R. Rees, J. Schünemann, W. Taxer, and G. Hawlitschek for the technical support in the selection and purification of SUMO[Eu]/SENP[Eu] variants; U. Plessman and H. Urlaub for the mass spectrometry analysis; and B. Schwappach (University Medical Center, Göttingen, Germany) for sharing yeast strains. We also thank M. Sola-Colom, T. Huyton, M. Aksu, and C. Paz for the critical reading of the manuscript.

This work was supported by the Max-Planck-Gesellschaft and the Deutsche Forschungsgemeinschaft (SFB860/B03).

The authors are inventors on a European patent application encompassing the here-described engineered SUMO/protease system. The authors declare no further competing financial interests.

Author contributions: A. Vera Rodriguez, S. Frey, and D. Görlich designed and performed experiments and analyzed and interpreted data; A. Vera Rodriguez and D. Görlich wrote the manuscript; and all authors approved the manuscript.

Submitted: 18 December 2018

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
