## [Reviewer comments · The Journal of Cell Biology]

Engineered SUMO/protease system identifies Pdr6 as a bidirectional nuclear transport receptor

Arturo Vera-Rodriguez, Steffen Frey, and Dirk Görlich

Corresponding Author(s): Dirk Görlich, Max Planck Institute for Biophysical Chemistry

Review Timeline:	Submission Date:	2018-12-18
	Editorial Decision:	2019-02-07
	Revision Received:	2019-03-26

Monitoring Editor: Larry Gerace

Scientific Editor: Melina Casadio

Transaction Report:

DOI: 10.1083/jcb.201812091

February 7, 2019

Re: JCB manuscript #201812091

Prof. Dirk Görlich
Max Planck Institute for Biophysical Chemistry
Am Faßberg 11
Göttingen 37077
Germany

Dear Prof. Görlich,

Thank you for submitting your manuscript entitled "Bidirectional nuclear transport by Pdr6 discovered through an engineered SUMO/protease system". The manuscript was assessed by expert reviewers, whose comments are appended to this letter. You will see that Reviewers 1-2-3 co-reviewed this submission together with your manuscript "Structural basis for the nuclear import and export functions of the biportin Pdr6/Kap122" and we also recruited a SUMO expert to weigh in on this work (#4). As discussed previously via email, we sincerely apologize again for the delay in sending you our decision. We invite you to submit a revision if you can address the reviewers' key concerns, as outlined here.

You will see that, like for the companion paper, there was a largely enthusiastic response to your submission, both to the scientific advances presented and the quality of the data. The reviewers thought that the orthogonal SUMO/SUMO protease system (Ms #1) was valuable both as a tool itself, and a model with potential applications to other proteases or related systems. Concerns were noted, however, by Reviewer #4, who was uncertain whether this is best suited for the JCB. After examining your manuscripts and reviews, we concur with Reviewers #1-3, and believe that both of your manuscripts would be acceptable to the JCB with some changes as outlined below.

Based on internal discussions of the work and the nature of the advance, and based upon the referee feedback, we think that this manuscript should be presented in the JCB Tools format. In addition, I think that the large amount of data presented in the main body of the manuscript would diminish the effectiveness of this paper for the more general audience of the JCB (as it may have done for Reviewer #4). Therefore, I ask that you reorganize this manuscript to include a maximum of 6 figures in the main body of the paper, moving the rest to the Supplemental section. For example, it would be possible to move Figures 4, 5, 6 and 9 to Supplemental, although I will leave decisions on exact changes up to you [noting the maximum number of supplemental figures for JCB Tools papers is 5]. Aside from these issues, the reviewers have enumerated a large number of clarifications to the text that you should address in revised versions of the manuscript. I don't believe you need to do any additional experiments for these to be acceptable. The suggestion of Reviewer #1 on your examining FG hydrogel behavior of eEF2, while a valid experiment, doesn't seem essential for your main message. However, you will want to address this concern with changes in the text.

1) Text limits: Character count for Tools is < 40,000, not including spaces. Count includes title page, abstract, introduction, results, discussion, acknowledgments, and figure legends. Count does not include materials and methods, references, tables, or supplemental legends.

-- JCB Tools do not have a combined "Results and Discussion" section; the two sections are separate. Please be sure to revise the manuscript organization prior to resubmission.

2) Titles, eTOC: Please consider the following revision suggestions aimed at increasing the accessibility of the work for a broad audience and non-experts.

Title: An engineered SUMO/protease system for eukaryotic expression demonstrates bidirectional nuclear transport by Pdr6

Running title (we can edit for you in the system if you run into any issue due to the character count): eukaryotic SUMO/protease system identifies Pdr6 as biportin
(we feel that highlighting the technological advance is also important for the running title and look forward to your thoughts on this point)

eTOC summary: A 40-word summary that describes the context and significance of the findings for a general readership should be included on the title page. The statement should be written in the present tense and refer to the work in the third person.

Please provide an eTOC statement at resubmission; please let us know if you would like any suggestions. We would encourage you to describe the technological and biological advances in this statement.

3) Figure formatting:

Molecular weight or nucleic acid size markers must be included on all gel electrophoresis. Please add molecular weight with unit labels on the following panels: 2D, 3ACD, 5A, 6A

4) Statistical analysis: Error bars on graphic representations of numerical data must be clearly described in the figure legend. The number of independent data points (n) represented in a graph must be indicated in the legend. Statistical methods should be explained in full in the materials and methods. For figures presenting pooled data the statistical measure should be defined in the figure legends.

5) Materials and methods: Should be comprehensive and not simply reference a previous publication for details on how an experiment was performed. Please provide full descriptions in the text for readers who may not have access to referenced manuscripts.

- Please be sure to provide all necessary information for others to attempt these experiments again, even if published in other works previously (e.g., for bacterial protein expression and purification methods)

- Microscope image acquisition: The following information must be provided about the acquisition and processing of images:

a. Make and model of microscope

b. Type, magnification, and numerical aperture of the objective lenses

c. Temperature

d. imaging medium

e. Fluorochromes

f. Camera make and model

g. Acquisition software

h. Any software used for image processing subsequent to data acquisition. Please include details and types of operations involved (e.g., type of deconvolution, 3D reconstitutions, surface or volume rendering, gamma adjustments, etc.).

6) A summary paragraph of all supplemental material should appear at the end of the Materials and methods section.

7) Conflict of interest statement: JCB requires inclusion of a statement in the acknowledgements regarding competing financial interests. If no competing financial interests exist, please include the following statement: "The authors declare no competing financial interests." If competing interests are declared, please follow your statement of these competing interests with the following statement: "The authors declare no further competing financial interests."

8) Author contributions: A separate author contribution section is required following the Acknowledgments in all research manuscripts. All authors should be mentioned and designated by their full names. We encourage use of the CRediT nomenclature.

9) Please resubmit individual, high-resolution files for the figures and editable individual files for the manuscript text and tables at resubmission.

GENERAL GUIDELINES:

Figures: Tools may have up to 10 main text figures. Figures must be prepared according to the policies outlined in our Instructions to Authors, under Data Presentation, <http://jcb.rupress.org/site/misc/ifora.xhtml>. Tools may have up to 5 supplemental figures. Up to 10 supplemental videos or flash animations are allowed. All figures in accepted manuscripts will be screened prior to publication.

*****IMPORTANT:** It is JCB policy that if requested, original data images must be made available. Failure to provide original images upon request will result in unavoidable delays in publication. Please ensure that you have access to all original microscopy and blot data images before submitting your revision.*******

The typical timeframe for revisions is three months; if submitted within this timeframe, novelty will not be reassessed at the final decision. Please note that papers are generally considered through only one revision cycle, so any revised manuscript will likely be either accepted or rejected.

Thank you for this interesting contribution to the Journal of Cell Biology. You can contact us at the journal office with any questions, cellbio@rockefeller.edu or call (212) 327-8588.

Sincerely,

Larry Gerace, PhD
Monitoring Editor, Journal of Cell Biology

Reviewer #1 (Comments to the Authors (Required)):

The SUMO--SUMO protease system for facile protein tagging and fast tag removal is of great advantage in the purification of heterologously expressed proteins, primarily in *E. coli*. The SUMO tag is generally soluble and unobtrusive, whereas its cleavage is orders of magnitude faster and more specific than any of the other proteases commonly used in tagging (e.g. thrombin, 3C proteases, TEV, etc.). Unfortunately, the SUMO--SUMO protease system has, thus far, only been effective in prokaryotic expression systems, thanks to their lack of endogenous SUMOylation as a post-translational modification. Indeed, attempts to use SUMO as a tag in eukaryotic organisms such as *S. cerevisiae* are thwarted by the rapid action of the endogenous machinery. To make things worse, but unsurprisingly given the ubiquity of SUMOylation as a regulatory post-translational modification in various cellular processes, the overexpression of SUMO-tagged proteins is often toxic to the eukaryotic cell. To make SUMO available as an efficient fusion-tag for recombinant expression in eukaryotic systems, Vera-Rodriguez et al. co-evolved an orthogonal SUMO and SUMO-protease system applicable in *S. cerevisiae* and mammalian cell lines. The authors engineered a resistance-dependent selection system to screen for SUMO and SUMO-protease variants that are resistant towards *S. cerevisiae* and human SUMO-proteases but instead are efficiently cleaved by the co-evolved SUMO-protease. To validate their candidates, the authors performed expression trials in *S. cerevisiae* and HEK293T cells to demonstrate stability of their SUMO-fusions towards endogenous SUMO-proteases, and, subsequent, efficient cleavage by the co-evolved protease. Moreover, the authors employed the developed system to identify cargo for the *S. cerevisiae* nuclear transport factor Pdr6. Interestingly, they discovered that Pdr6 binds to both import and export cargo in the absence and presence of RanGTP, respectively, concluding that Pdr6 can facilitate bidirectional transport. Thus, the authors introduce the term "biportin" for nuclear transport factors with bidirectional functionality. Together, the current manuscript elegantly demonstrates the development of a valuable tool for recombinant protein expression in eukaryotic cells employing the SUMO-fusion system. The developed method is an important advance in the field. To make the manuscript more accessible to the broad readership of JCB, the authors should address the following specific points:

1. Figure 5A shows a SUMO-Citrine system, the corresponding text describes SUMO-GFP. This section requires clarification.
2. Figure 3A demonstrates the (absence of) cleavage of SUMO^{Eu} variants by co-evolved SENP variants. According to the figure legend, the experiment was performed at 100 μ M SUMO-MBP and 20 nM protease. Comparison of SENP^{Eu}B and SUMO^{Eu}1 in Figure 3A with Figure 3D at a protease concentration of 20 nM suggests different experimental parameters. Whereas the ratio of cleaved to uncleaved SUMO^{Eu}1 should be identical between the two experiments, the data in Figure 3A suggest a protease concentration of less than 10 nM. The authors should explain these differences.
3. The authors state: "The data illustrate that all SUMO^{Eu} variants [...] are highly resistant against scUlp1, the SUMOstar protease and hsSENP2. The SUMO^{Eu}1 fusion got, however, rapidly cleaved by five of the six selected SENP^{Eu} proteases (B, G, H, J, K)." When comparing the ratio between

cleaved and uncleaved SUMO-MBP between hsSEN2 and SENPEuK, the statement "highly resistant" versus "rapidly cleaved" does not appear appropriate. The text should be revised accordingly.

4. In wildtype cells eEF2 is excluded from the nucleus, whereas in Δ Pdr6 cells eEF2 shows nuclear accumulation despite its larger size of ~93 kDa (~120 kDa as GFP fusion). The authors speculate about the unexpected nuclear localization of eEF2 and one of their explanations focuses on the "translation-friendly side chains that should accelerate leakage into nuclei". In light of the recent publication from the Gorlich group highlighting those "translation-friendly" residues, the authors should employ their FG-hydrogel partitioning experiment to validate their hypothesis.

5. In the SDS-PAGE readouts of SUMO cleavage reactions (e.g. Fig. 2D) it would be preferable to also show the portion of the gel where SUMO migrates, if the gels are available.

6. There are several instances of "data not shown"; it would be more appropriate to show the data in the supplementary figures.

7. On p. 11, the following statement is unclear within its context: "Without further addition, RanGAP will dominate and create a cytoplasmic (low RanGTP) environment that favors binding of import substrates. To mimic also a nuclear environment, one sample was supplemented with His-tagged RanGTP (added as a GTPase-locked mutant)." It seems that the authors are referring to what happens in cell free lysate, but that is not explicitly stated. The text should be revised accordingly.

8. On p. 12, the argument is made that "Binding assays with only recombinant components and immobilized Ubc9 confirmed the Ubc9-Pdr6 interaction as being specific and direct." Whereas the experiment shows that the interaction is direct, no explicit hypothesis about "specificity" is either posed or tested. Please include additional data that supports this statement or revise text accordingly.

9. English and Grammar. The manuscript would benefit from editing by the senior author.

Reviewer #2 (Comments to the Authors (Required)):

The two manuscripts by Asku et al and Vera-Rodriguez et al define yeast Prd6 as a bidirectional NTR and provide the structural basis for its cargo recognition. Each of the two manuscripts is experimentally very extensive and taken together they define novel yeast export pathway that was previously unknown.

Major comments on Vera-Rodriguez et al:

The authors have developed an elegant readout for the specificity of mutagenized SUMO proteases towards mutagenized SUMO tags and used it to evolve SUMO tag / SUMO protease pairs that resist cleavage of endogenous SUMO proteases in human and yeast cells. The authors demonstrate that their system is superior to existing sumo affinity tags, in terms of cleavage kinetics and stability, even in human cells. The breadth of this screen is impressive and I believe it has good chances to make a high impact, not only because of its immediate outcome but also because it might be adapted to other protease substrate pairs.

The authors employ their system to identify cargos of yeast Prd6. They validate ubc9 as an import,

as well as Phil1, Lsp1, eIF5A, eEF2 as export cargos. The experiments presented are of very high quality and convincing. The concept of bidirectional active nuclear transport is per se not novel, and the authors cite the respective papers on Imp13, etc. in the introduction. This capacity however is newly identified for Prd6 by this study, which is a very exciting finding that broadens our understanding of the nuclear transport machinery.

I most enthusiastically recommend this paper for publication in JCB after minor revisions.

Minor comments on Vera-Rodriguez et al:

- Fig. 1A: This would be more straight forward to comprehend if the cartoon would include two arrows pointing to reaction products and their activity in case PCSfor or PCSagainst are cleaved. You might also indicate the degron that uses the N-end rule in the cartoon.
- Along similar lines, a flow chart illustrating the conceptual layout of the screen would be helpful. The authors have included very few supplementary items thus far and still have space.
- It took me ages to figure out that the numbers shown on top of the lanes in Figure 3A correspond to ones shown in Figure 3B in tiny fonts size. Please consider making this more user friendly, e.g. mention in (A) that this is explained in (B), or better swap both panels.
- Regarding the term 'biportin', I am not sure if suggesting yet another term is a good approach for reducing the confusion about nomenclature in this field. What is a transportin than? Would one have to rename Imp13, Xpo4, Xpo7 and Msn5? But have it your way ...
- Discussion: A model in which expelling translation factors from the nucleus is important to prevent nuclear translation generously assumes that pre-ribosomes (when given a chance) would have translational activity. But is this really the case? Even if so, expelling initiation factors would be sufficient to prevent nuclear translation and it is not clear why there should be evolutionary pressure on all translation factors. In that sense, the authors explanation of 'metabolic burden of mis-localized proteins' or maybe better 'kinetic burden' of reduced concentrations of translation factors in the cytosol, is a simpler and in my view also more plausible interpretation.

Major comments on Asku et al:

The authors have solved structures of key intermediates of the Prd6 transport cycle, specifically of ubc9-Prd6 import complex, RanGTP-Prd6 without cargo and RanGTP-Prd6-eIF5A export complex. The structures show some differences to their metazoan counterparts, that is the respective Xpo4 and Imp13 structures. These structures reveal the molecular basis for both Prd6-depedent pathways, as defined in the accompanying paper, in very high detail. The unconventional, bidirectional transport scheme of these pathways make this a very interesting paper, that does considerably go beyond our present structural knowledge of NTRs and their substrate binding mode. The fact that Prd6 also covers key interfaces of its substrates suggesting a chaperoning function is highly interesting! On a general note, solving as many cargo-NTR complexes as possible is important for this field. Since there is such a huge number of them, the existing structural repertoire is still quite limited.

I most enthusiastically recommend this paper for publication in JCB after minor revisions.

Minor comments on Asku et al:

- Page 6: Please make clear in the main text if the 'complex of full-length Pdr6 and the GTPase-deficient RanQ69L' contains yeast or human Ran. Does not matter much since its super-conserved, but should be transparent. Likewise, please clarify for Prd6-RanGTP complex.
- Page 7: Regarding the statement 'Furthermore, we found that Pdr6 imports UBC9, exports eIF5A and thus combines Imp13 and Xpo4 functions. This could suggest that Imp13 and Xpo4 evolved by gene duplication and diversification from an ancestral Pdr6 gene.' If you first say 'gene duplication

and diversification', you make one assume that the ubc9 and eIF5A binding function existed before gene duplication, which would be inconsistent with that function having 'appeared twice in evolution'. What you mean to say is that most likely an ancient NTR gene was duplicated and the resulting genes adopted ubc9 and eIF5A binding functions independently, correct? Please consider making this clearer.

On another note, the simplest imaginable evolutionary trajectory is not necessarily the most likely. In principle it could be both, convergent or divergent evolution, right?

Comment on both papers:

Why would Phil1, Lsp1, eIF5A and eEF2 be exported by a different pathway than crm1-dependent export?! The authors suggest that an NES might interfere with the activity of a 'highly optimized translation factor', however, this argument does not apply to all Prd6 cargos. Could this have to do with the proposed nuclear chaperoning function of i.e. the hypusine-containing interface? But what is the evidence really that exposing this interface in the nucleus is harmful? Can one experimentally test this idea? Would exposing this interface in the nucleus by overexpressing eIF5A fused to a classical NLS (on top of the endogenous protein) cause a reduced fitness phenotype that depends on the expression level? Likewise, several other hypotheses spelled out in the discussion of mainly Asku et al, could well be tested. In more general terms, the strength of both papers is that they provide a very solid base to test the physiological relevance of the Prd6-dependent pathways. The two papers however do not yet capitalize on this capacity, which would be quite interesting because of the relevance for metazoan Xpo4 and Imp13. Both papers however already comprise an experimental tour de force. Extending the scope even further is certainly not required for publication.

Reviewer #3 (Comments to the Authors (Required)):

This manuscript evolves cleaving tags from SUMO/SEN1 system. This authors' group have previously developed a tag-cleaving protease system for purifying recombinant proteins. Although the system is efficient when used on recombinant proteins, the system is not suitable for use in eukaryotic cellular system, because endogenous SUMO protease cleaves SUMO tag prematurely. To overcome this problem, the authors evolved SUMO/SEN1 systems, through their original assay, that have novel specificities; SUMO variants (termed, Eu) that resist cleavage by animal and yeast SUMO protease, SEN1 proteases (termed, Eu) that cleave the evolved SUMO variants (Eu). They used these tags for affinity-capture-and proteolytic release approach, and identified new Pdr6 (importin beta family nuclear transport receptor) cargoes. Addition to new import cargo, they also identified new export cargoes, which means Pdr6 mediates both nuclear import and nuclear export.

Overall experiments are well done, especially, the way they evolved new SUMO variants and SEN1 proteases are based on their original strategy, which is very sagacious. The identification of cargoes of Pdr6, which showed this NTR mediates bidirectional transport is also interesting. I just wondered, what is the merit for including first affinity chromatography and SEN1 (EuB) elution in Fig 7? Only His affinity chromatography and 3M Gdn-HCl elution do not yield good results? Please describe the strength of the use of new tag, if possible.

Minor Comments:

Description of SUMO (Eu, Eu1), SEN1 (Eu, EuB..) in the text is abrupt. I understood after I saw the Fig 3B.

Reviewer #4 (Comments to the Authors (Required)):

The manuscript "Bidirectional nuclear transport by Pdr6 discovered through an engineered SUMO/protease system" by Vera-Rodriguez et al consists of three parts:

The large part of the manuscript (Figures 1-7) describes a biotechnological advancement of SUMO as a fusion tag that can be selectively cleaved by a specific SUMO protease: based on published structures of SUMO/Senp pairs, the authors permutate specific residues in all possible combinations and use an elegant viability screen with a modified antibiotic resistance gene that allows selection of the desired novel pairs of SUMO and protease variants. Through this approach, the authors identify a number of SUMO variants that can only be cleaved by specific variants of a SUMO protease.

The design of the screen is appealing and the data are technically sound. In the long-run, these novel pairs of SUMO proteins and proteases may indeed be very useful, e.g. to cleave SUMO in cells at will with an inducible protease variant. This is however something the authors do not discuss. Rather, they point out that the novel SUMO-fusion proteins are stable in eukaryotic extracts. But NEM, iodoacetamide or chemically modified SUMO variants that function as suicide inhibitors would have done the same trick.....

From a technological perspective, as a proof of concept of how to select protease/substrate variants, I find this part of the manuscript quite interesting and certainly suitable for a biotechnology journal.

In the second part of the manuscript (starting with Figure 7) the authors use one of their novel SUMO/protease pairs to identify novel binding partners for the *S. cerevisiae* nuclear transport receptor Pdr6/Kap122 by pulldowns from yeast extracts. They do this either in the absence or presence of RanGTP, which disrupts import complexes but stabilizes export complexes. These experiments lead to the identification of several known and novel interaction partners.

However, there is no reason to believe that one could not have found the same interaction partners with any other available double affinity tag approach. And in contrast to several other recent proteomics publications, the data presented here do not provide a comprehensive data set of the Pdr6 interactome.

One novel binding partner is Ubc9, which binds specifically Pdr6, and no other yeast transport receptor (Figure 8). Delta Pdr6 cells miss-localize Ubc9-GFP to the cytoplasm (Figure 8D). Endogenous Ubc9 was not tested.

Is this interaction biologically particularly important and thus interesting? Is the binding modus interesting? What is the NLS? How similar is Pdr6 to importin 13, which binds mammalian Ubc9? The data shown in the current manuscript do not address this. Instead, there seems to be an accompanying manuscript which describes the interaction of Pdr6 and Ubc9 including crystal structure.

Finally, in Figures 9 and 10 the authors present some evidence that Pdr6 not only acts as an import receptor but can also act as an export receptor. This is a recurring observation that the authors now have made and published several times (e.g. for the mammalian transport receptor importin 13

in EMBO 2001, and for Xpo7 in JCB 2018). For the candidates Lsp1 and Pil1, only pulldowns are shown, biological relevance is lacking. For eIF5A, pulldown and partial miss-localization are shown, but again, only GFP-tagged eIF5A was shown.

Taken together, the technically sound and well-written manuscript has a biotechnologically interesting part that may or may not be interesting for the JCB community. The nucleocytoplasmic transport aspects are minimalistic. They are fine to validate the novel SUMO fusion tag, but lack in itself the depth to be biologically particularly interesting - depth is instead provided in an independent manuscript. In consequence, I am not really convinced that this manuscript is sufficient for JCB.

Max-Planck-Institut für Biophysikalische Chemie

Dr. Dirk Görlich · Managing Director

Am Fassberg 11 · D-37077 Göttingen · E-mail: goerlich@mpibpc.mpg.de · Tel.: ++49 551 2012401

To the Editors of The Journal of Cell Biology
- via the internet -

Göttingen, 26th of March 2019

JCB manuscripts #201812091 and #201812093

Dear Editor, dear Reviewers,

Thank you very much for the very positive evaluation of our two manuscripts and the valuable input for further improvements. We have prepared accordingly revised versions, which we hope can now be accepted for publication. A list of changes and answers to all points raised by the reviewers follow on the next pages. We combined both replies into one document.

With best regards,

Dirk Görlich

JCB #201812091 (Vera-Rodriguez, Frey & Görlich).

Major changes to the manuscript:

- We added an eTOC summary
- We added a running title
- We changed the article to a ‘Tools’ format, which included moving four main figures to the supplements and separating results from discussion.
- We expanded the Methods section and implemented a number of smaller changes as detailed in the point-to-point reply below.

Reviewer #1 (We repeated the comments in blue in front of each of our answers)

The SUMO--SUMO protease system for facile protein tagging and fast tag removal is of great advantage in the purification of heterologously expressed proteins, primarily in *E. coli*. The SUMO tag is generally soluble and unobtrusive, whereas its cleavage is orders of magnitude faster and more specific than any of the other proteases commonly used in tagging (e.g. thrombin, 3C proteases, TEV, etc.). Unfortunately, the SUMO--SUMO protease system has, thus far, only been effective in prokaryotic expression systems, thanks to their lack of endogenous SUMOylation as a post-translational modification. Indeed, attempts to use SUMO as a tag in eukaryotic organisms such as *S. cerevisiae* are thwarted by the rapid action of the endogenous machinery. To make things worse, but unsurprisingly given the ubiquity of SUMOylation as a regulatory post-translational modification in various cellular processes, the overexpression of SUMO-tagged proteins is often toxic to the eukaryotic cell. To make SUMO available as an efficient fusion-tag for recombinant expression in eukaryotic systems, Vera-Rodriguez et al. co-evolved an orthogonal SUMO and SUMO-protease system applicable in *S. cerevisiae* and mammalian cell lines. The authors engineered a resistance-dependent selection system to screen for SUMO and SUMO-protease variants that are resistant towards *S. cerevisiae* and human SUMO-proteases but instead are efficiently cleaved by the co-evolved SUMO-protease. To validate their candidates, the authors performed expression trials in *S. cerevisiae* and HEK293T cells to demonstrate stability of their SUMO-fusions towards endogenous SUMO-proteases, and, subsequent, efficient cleavage by the co-evolved protease. Moreover, the authors employed the developed system to identify cargo for the *S. cerevisiae* nuclear transport factor Pdr6. Interestingly, they discovered that Pdr6 binds to both import and export cargo in the absence and presence of RanGTP, respectively, concluding that Pdr6 can facilitate bidirectional transport. Thus, the authors introduce the term "biportin" for nuclear transport factors with bidirectional functionality. Together, the current manuscript elegantly demonstrates the development of a valuable tool for recombinant protein expression in eukaryotic cells employing the SUMO-fusion system. The developed method is an important advance in the field. To make the manuscript more accessible to the broad readership of JCB, the authors should address the following specific points:

1. Figure 5A shows a SUMO-Citrine system, the corresponding text describes SUMO-GFP. This section requires clarification.

This was indeed inconsistent. We have now written ‘YFP’ throughout the paper, the rationale being that Citrine is codon-optimized YFP, with YFP being the more common term. The problem arose, because we used an anti-GFP nanobody as an interaction partner. This is now described as an anti-YFP/GFP with a reference given (Kirchhofer et al., 2010).

2. Figure 3A demonstrates the (absence of) cleavage of SUMO^{Eu} variants by co-evolved SENP variants. According to the figure legend, the experiment was performed at 100 μ M SUMO-MBP and 20 nM protease. Comparison of SENP^{EuB} and SUMO^{Eu1} in Figure 3A with Figure 3D at a protease concentration of 20 nM suggests different experimental parameters. Whereas the ratio of cleaved to uncleaved SUMO^{Eu1} should be identical between the two experiments, the data in Figure 3A suggest a protease concentration of less than 10 nM. The authors should explain these differences.

The SENP^{EuB} experiment in the former Figure 3D was an outlier with an unusually high activity, suggesting an experimental error. It has been removed. Thank very much for pointing out this issue.

3. The authors state: "The data illustrate that all SUMO^{Eu} variants [...] are highly resistant against scUlp1, the SUMOstar protease and hsSENP2. The SUMO^{Eu1} fusion got, however, rapidly cleaved by five of the six selected SENPEu proteases (B, G, H, J, K)." When comparing the ratio between cleaved and uncleaved SUMO-MBP between hsSENP2 and SENPEuK, the statement "highly resistant" versus "rapidly cleaved" does not appear appropriate. The text should be revised accordingly.

Please note that we used scUlp1, the SUMOstar protease and hsSENP2 at 50-fold higher concentrations than the SENPEu proteases. This factor needs to be considered for estimating turnover rates. SENPEu^H is thus ≥ 1000 times more active on a SUMO^{Eu1} substrate than the reference proteases.

4. In wildtype cells eEF2 is excluded from the nucleus, whereas in Δ Pdr6 cells eEF2 shows nuclear accumulation despite its larger size of ~93 kDa (~120 kDa as GFP fusion). The authors speculate about the unexpected nuclear localization of eEF2 and one of their explanations focuses on the "translation-friendly side chains that should accelerate leakage into nuclei". In light of the recent publication from the Gorlich group highlighting those "translation-friendly" residues, the authors should employ their FG-hydrogel partitioning experiment to validate their hypothesis.

This is a great experiment, but a project on its own. EF2 does not fold properly when expressed in *E. coli*, so we would have to produce the mutants by eukaryotic expression. Furthermore, as there are multiple 'translocation-friendly' residues, we would have to simultaneously introduce multiple mutations. The proof that folding was not affected would then become a serious issue, in particular as tests for proper folding would not be as straightforward as in the quoted GFP-engineering study.

5. In the SDS-PAGE readouts of SUMO cleavage reactions (e.g. Fig. 2D) it would be preferable to also show the portion of the gel where SUMO migrates, if the gels are available.

We do have these gels (see example below). However, showing these gels *in toto* will require four times more space and make it impossible to fit the dataset onto one page. This applies in particular to Figure 3. There is no gain of information for the reader. Furthermore, we have already exhausted our slots for supplementals, and thus cannot show the complete gels to the readers.

6. There are several instances of "data not shown"; it would be more appropriate to show the data in the supplementary figures.

With one table and five (multi-panel) supplemental figures, we are already exceeding our limit for supplementals and cannot add more data. However, on looking through the text again, it became clear that none of the "data not shown" statements is actually needed. The four instances have now been resolved as follows:

"...the ribosome-binding site and obtained optimal results with a weaker one with a ~50-fold reduced translation-efficiency (data not shown)."

We now list sequences of the original RBS and the attenuated one in the Methods section.

"...into the sensor plasmid and selected against cleavage by the SUMOstar protease, the so far most promiscuous SUMO-cleaving enzyme (data not shown but see also below)."

We were referring to the fact that the SUMOstar protease cleaves not only SUMOstar substrates, but also wild type SUMO-fusions. The fact is actually evident from the literature (Peroutka et al., 2008) and Figure 3B (former Figure 3A). We re-phrased the sentence accordingly.

"The SENP^{Eu} variants survive ≥ 30 freeze-thaw cycles without a noticeable loss in activity (data not shown)."

This statement was meant as a technical advice. It actually also applies to the parental wild type bdSEN1 protease. In fact, this has become a standard test for our lab-produced enzymes, also for assessing whether a given storage buffer is appropriate. Showing data would be a bit trivial. We therefore removed the sentence.

"The SENP^{Eu} B, H, and K variants cut wild-type scSUMO fusions ~1000-fold less efficiently than their preferred substrate (Fig.3A, and data not shown)."

This statement is already fully supported by the quoted Figure 3A (now 3B).

7. On p. 11, the following statement is unclear within its context: "Without further addition, RanGAP will dominate and create a cytoplasmic (low RanGTP) environment that favors binding of import substrates. To mimic also a nuclear environment, one sample was supplemented with His-tagged RanGTP (added as a GTPase-locked mutant)." It seems that the authors are referring to what happens in cell free lysate, but that is not explicitly stated. The text should be revised accordingly.

Yes, we did refer to a binding reaction in a yeast extract. We modified the text accordingly.

8. On p. 12, the argument is made that "Binding assays with only recombinant components and immobilized Ubc9 confirmed the Ubc9-Pdr6 interaction as being specific and direct." Whereas the experiment shows that the interaction is direct, no explicit hypothesis about "specificity" is either posed or tested. Please include additional data that supports this statement or revise text accordingly.

The (positive) binding assay with recombinant components indeed demonstrates a direct interaction, and it rules out that we missed some bridging factor. In Figure 5B, however, we also tested specificity, the outcome being that UBC9 bound Pdr6 but no other NTRs (Lph2, Mtr10, Msn5, Nmd5, Yrb4, Smx1, importin α and importin β and thus all known importers have been tested). We modified the text to make this clearer.

9. English and Grammar. The manuscript would benefit from editing by the senior author.

The manuscript has been proofread and corrected again.

Reviewer #2

The two manuscripts by Asku et al and Vera-Rodriguez et al define yeast Prd6 as a bidirectional NTR and provide the structural basis for its cargo recognition. Each of the two manuscripts is experimentally very extensive and taken together they define novel yeast export pathway that was previously unknown.

Major comments on Vera-Rodriguez et al:

The authors have developed an elegant readout for the specificity of mutagenized SUMO proteases towards mutagenized SUMO tags and used it to evolve SUMO tag / SUMO protease pairs that resist cleavage of endogenous SUMO proteases in human and yeast cells. The authors demonstrate that their system is superior to existing sumo affinity tags, in terms of cleavage kinetics and stability, even in human cells. The breadth of this screen is impressive and I believe it has good chances to make a high impact, not only because of its immediate outcome but also because it might be adapted to other protease substrate pairs.

The authors employ their system to identify cargos of yeast Prd6. They validate *ubc9* as an import, as well as *Phil1*, *Lsp1*, *eIF5A*, *eEF2* as export cargos. The experiments presented are of very high quality and convincing. The concept of bidirectional active nuclear transport is per se not novel, and the authors cite the respective papers on *Imp13*, etc. in the introduction. This capacity however is newly identified for Prd6 by this study, which is a very exciting finding that broadens our understanding of the nuclear transport machinery.

I most enthusiastically recommend this paper for publication in JCB after minor revisions.

Thank you!

Minor comments on Vera-Rodriguez et al:

- Fig. 1A: This would be more straight forward to comprehend if the cartoon would include two arrows pointing to reaction products and their activity in case PCSfor or PCSagainst are cleaved. You might also indicate the degron that uses the N-end rule in the cartoon.

We re-designed the figure which now not only shows the intact sensor but also its cleavage products and their stabilities.

- Along similar lines, a flow chart illustrating the conceptual layout of the screen would be helpful. The authors have included very few supplementary items thus far and still have space.

By reformatting the manuscript from an article to a tools format, we have already reached the limit for supplements. However, we addressed this point already by extending Figure 1.

- It took me ages to figure out that the numbers shown on top of the lanes in Figure 3A correspond to ones shown in Figure 3B in tiny fonts size. Please consider making this more user friendly, e.g. mention in (A) that this is explained in (B), or better swap both panels.

We have swapped the panels as suggested and increased the font sizes of the superscripts.

- Regarding the term 'biportin', I am not sure if suggesting yet another term is a good approach for reducing the confusion about nomenclature in this field. What is a transportin than? Would one have to rename *Imp13*, *Xpo4*, *Xpo7* and *Msn5*? But have it your way ...

We did not rename any protein. Instead, we propose the new term “biportin” as a functional category, to complement the terms “importin” and “exportin”.

In contrast, “transportin” is a protein name, given with the assumption that it mediates not only import of hnRNP proteins but also export of mRNA. Now we know that the latter assumption is incorrect, which also implies that the name is “burned” as a generic name for a bidirectional transporter.

- Discussion: A model in which expelling translation factors from the nucleus is important to prevent nuclear translation generously assumes that pre-ribosomes (when given a chance) would have

translational activity. But is this really the case? Even if so, expelling initiation factors would be sufficient to prevent nuclear translation and it is not clear why there should be evolutionary pressure on all translation factors. In that sense, the authors explanation of 'metabolic burden of mis-localized proteins' or maybe better 'kinetic burden' of reduced concentrations of translation factors in the cytosol, is a simpler and in my view also more plausible interpretation.

We certainly do not assume that pre-ribosomes have any translation activity. Nevertheless, there are (still!) highly cited and yet fundamentally flawed papers that argue that 30% of all translation occur inside nuclei (*see: Iborra et al., 2001. Coupled transcription and translation within nuclei of mammalian cells. Science. 293:1139-1142.*) Therefore, it is important to make the point that nuclear translation is actively suppressed.

We do agree that all the individual measures of a cell to suppress nuclear translation are probably an overkill. Nevertheless, this might also be an issue of robustness. A damage of the nuclear envelope (or open mitosis in metazoans) would give all translation components (including mature ribosomes) access to the nuclear interior. It then makes sense that all of them get actively retrieved. Likewise, it is well possible that mis-localization of (the typically RNA-binding) translation factors impairs nuclear functions by competing with cognate interactions. We are assuming that all these facets contribute to the evident evolutionary pressure for translation factor retrieval to the cytoplasm. We think that our discussion is well balanced in this respect.

Reviewer #3

This manuscript evolves cleaving tags from SUMO/SEN1 system. This authors' group have previously developed a tag-cleaving protease system for purifying recombinant proteins. Although the system is efficient when used on recombinant proteins, the system is not suitable for use in eukaryotic cellular system, because endogenous SUMO protease cleaves SUMO tag prematurely. To overcome this problem, the authors evolved SUMO/SEN systems, through their original assay, that have novel specificities; SUMO variants (termed, Eu) that resist cleavage by animal and yeast SUMO protease, SENP proteases (termed, Eu) that cleave the evolved SUMO variants (Eu). They used these tags for affinity-capture-and proteolytic release approach, and identified new Pdr6 (importin beta family nuclear transport receptor) cargoes. Addition to new import cargo, they also identified new export cargoes, which means Pdr6 mediates both nuclear import and nuclear export.

Overall experiments are well done, especially, the way they evolved new SUMO variants and SENP proteases are based on their original strategy, which is very sagacious. The identification of cargoes of Pdr6, which showed this NTR mediates bidirectional transport is also interesting. I just wondered, what is the merit for including first affinity chromatography and SENP (EuB) elution in Fig 7? Only His affinity chromatography and 3M Gdn-HCl elution do not yield good results? Please describe the strength of the use of new tag, if possible.

The background of a direct Ni(ii) chelate chromatography is just too high, because yeast and human cells contain numerous proteins with oligo-histidine tags. This point actually becomes clear with the experiment where we expressed and purified the actin-mutant in HEC293T cells.

Minor Comments:

Description of SUMO (Eu, Eu1), SENP (Eu, EuB..) in the text is abrupt. I understood after I saw the Fig 3B.

We re-wrote this part of the text.

Reviewer #4

The manuscript "Bidirectional nuclear transport by Pdr6 discovered through an engineered SUMO/protease system" by Vera-Rodriguez et al consists of three parts:

The large part of the manuscript (Figures 1-7) describes a biotechnological advancement of SUMO as a fusion tag that can be selectively cleaved by a specific SUMO protease: based on published structures of SUMO/Senp pairs, the authors permute specific residues in all possible combinations and use an elegant viability screen with a modified antibiotic resistance gene that allows selection of the desired novel pairs of SUMO and protease variants. Through this approach, the authors identify a number of SUMO variants that can only be cleaved by specific variants of a SUMO protease.

The design of the screen is appealing and the data are technically sound. In the long-run, these novel pairs of SUMO proteins and proteases may indeed be very useful, e.g. to cleave SUMO in cells at will with an inducible protease variant. This is however something the authors do not discuss.

This has been discussed as follows:

"This opens interesting experimental avenues, such as a broader re-engineering of the SUMO system in living cells, conditionally subjecting (SUMO^{Eu}-tagged) proteins to degradation by the N-end-rule pathway, or detaching them from an anchor site or transport signal. So far, TEV-protease has been the first choice for the latter applications (see e.g. Taxis and Knop, 2012). With the far more active SENP^{EuB} protease, however, we expect faster and more complete responses."

Rather, they point out that the novel SUMO-fusion proteins are stable in eukaryotic extracts. But NEM, iodoacetamide or chemically modified SUMO variants that function as suicide inhibitors would have done the same trick..... From a technological perspective, as a proof of concept of how to select protease/substrate variants, I find this part of the manuscript quite interesting and certainly suitable for a biotechnology journal.

NEM or iodoacetamide are generic thiol-reactive agents and therefore modify and kill many other proteins, prominent examples being NSF (NEM-sensitive fusion factor) or NTRs (containing usually ≥ 20 largely exposed cysteines). Besides these examples, in complex lysates hundreds of proteins harboring exposed cysteines are prone to be modified by NEM or iodoacetamide. Frankly, I would not trust interaction assays that get biased by such rather non-specific protein-modification. The inhibitor-treatment also does not help when the protein of interest is actually produced in a eukaryotic cell (rather than added to a treated extract). We believe that recombinant protein production in eukaryotic hosts will be the most important application of our technology.

In the second part of the manuscript (starting with Figure 7) the authors use one of their novel SUMO/protease pairs to identify novel binding partners for the *S. cerevisiae* nuclear transport receptor Pdr6/Kap122 by pulldowns from yeast extracts. They do this either in the absence or presence of RanGTP, which disrupts import complexes but stabilizes export complexes. These experiments lead to the identification of several known and novel interaction partners.

However, there is no reason to believe that one could not have found the same interaction partners with any other available double affinity tag approach.

You will not find any such claim in our paper. Nevertheless, our strategy has a unique combination of advantages over previous technologies (e.g. in comparison to Tev protease shorter protease elution times at lower temperature, far less protease contaminating the eluates, etc). Likewise we wish to note that earlier attempts of mapping Pdr6-interactions failed in identifying Ubc9 and the export cargoes.

And in contrast to several other recent proteomics publications, the data presented here do not provide a comprehensive data set of the Pdr6 interactome.

This will come in a broader context. Concerning the current manuscript, there would not have been the space for a major expansion, and opening yet another chapter would compromise the readability of the paper.

One novel binding partner is Ubc9, which binds specifically Pdr6, and no other yeast transport receptor (Figure 8). Delta Pdr6 cells miss-localize Ubc9-GFP to the cytoplasm (Figure 8D). Endogenous Ubc9 was not tested.

We tagged Ubc9 genomically and would therefore regard it as an endogenous protein, in particular as the tagged version was the only copy of the (essential) *UBC9* gene, and the tagging caused no obvious phenotype.

Is this interaction biologically particularly important and thus interesting? Is the binding mode interesting? What is the NLS? How similar is Pdr6 to importin 13, which binds mammalian Ubc9? The data shown in the current manuscript do not address this. Instead, there seems to be an accompanying manuscript which describes the interaction of Pdr6 and Ubc9 including crystal structure.

Indeed, we addressed all these questions in the accompanying manuscript, which includes three structures, namely that of the Pdr6·Ubc9 complex, of the RanGTP·Pdr6 complex and the RanGTP·Pdr6·eIF5A complex.

Finally, in Figures 9 and 10 the authors present some evidence that Pdr6 not only acts as an import receptor but can also act as an export receptor. This is a recurring observation that the authors now have made and published several times (e.g. for the mammalian transport receptor importin 13 in EMBO 2001, and for Xpo7 in JCB 2018). For the candidates Lsp1 and Pil1, only pulldowns are shown, biological relevance is lacking. For eIF5A, pulldown and partial miss-localization are shown, but again, only GFP-tagged eIF5A was shown.

The reviewer missed that we also demonstrated nuclear export of EF2 by Pdr6. As to the criticism of having analyzed the localization of GFP-fusions: this is pretty much standard. In fact, GFP enlarges the protein, and one would expect the non-tagged protein to be actually even more sensitive to a loss of active nuclear export. The alternative of using antibodies for detecting the endogenous protein would also have issues: potential fixation artifacts and not knowing if the antibodies are specific for their targets (note that eIF5A and EF2 are both essential, so knockouts cannot serve as antibody specificity controls).

Taken together, the technically sound and well-written manuscript has a biotechnologically interesting part that may or may not be interesting for the JCB community. The nucleocytoplasmic transport aspects are minimalistic. They are fine to validate the novel SUMO fusion tag, but lack in itself the depth to be biologically particularly interesting - depth is instead provided in an independent manuscript. In consequence, I am not really convinced that this manuscript is sufficient for JCB.

We assume the real problem is that this reviewer has not seen the second paper. We hope that she/ he will enjoy the complete story once it has been published as two back-to-back papers.